



# Cloud adjustments from large-scale smoke-circulation interactions strongly modulate the southeast Atlantic stratocumulus-to-cumulus transition

Michael S. Diamond[1,2], Pablo E. Saide[3,4], Paquita Zuidema[5], Andrew S. Ackerman[6], Sarah J. Doherty[7,8], Ann M. Fridlind[6], Hamish Gordon[9], Calvin Howes[3], Jan Kazil[1,2], Takanobu Yamaguchi[1,2], Jianhao Zhang[1,2], Graham Feingold[2], and Robert Wood[8]

[1]Cooperative Institute for Research in Environmental Sciences (CIRES), University of Colorado, Boulder, Colorado, USA
[2]NOAA Chemical Sciences Laboratory (CSL), Boulder, Colorado, USA
[3]Department of Atmospheric and Oceanic Sciences, University of California, Los Angeles, CA USA
[4]Institute of the Environment and Sustainability, University of California, Los Angeles, CA, USA
[5]Rosenstiel School of Marine and Atmospheric Science, University of Miami, Miami, FL, USA
[6]NASA Goddard Institute for Space Studies, New York, NY, USA
[7]Cooperative Institute for Climate, Ocean and Ecosystem Studies, Seattle, WA, USA
[8]Department of Atmospheric Sciences, University of Washington, Seattle, WA, USA
[9]Engineering Research Accelerator and Center for Atmospheric Particle Studies, Carnegie Mellon University, Pittsburgh, PA, USA

*Correspondence to*: Michael S. Diamond (michael.diamond@noaa.gov)

**Abstract.** Smoke from southern Africa blankets the southeast Atlantic Ocean from June-October, producing strong and competing aerosol radiative effects. Smoke effects on the transition between overcast stratocumulus and scattered cumulus clouds are investigated along a Lagrangian (air-mass-following) trajectory in regional climate and large eddy simulation models. Results are compared with observations from three recent field campaigns that took place in August 2017: ORACLES, CLARIFY, and LASIC. The case study is set up around the joint ORACLES-CLARIFY flight that took place near Ascension Island on 18 August 2017. Smoke sampled upstream on an ORACLES flight on 15 August 2017 likely entrained into the marine boundary layer later sampled during the joint flight.

The case is first simulated with the WRF-CAM5 regional climate model in three distinct setups: 1) FireOn, in which smoke emissions and any resulting smoke-cloud-radiation interactions are included; 2) FireOff, in which no smoke emissions are included; and 3) RadOff, in which smoke emissions and their microphysical effects are included but aerosol does not interact directly with radiation. Over the course of the Lagrangian trajectory, differences in free tropospheric thermodynamic properties between FireOn and FireOff are nearly identical to those between FireOn and RadOff, showing that aerosol-radiation interactions are primarily responsible for the free tropospheric effects. These effects are non-intuitive: in addition to the expected heating within the core of the smoke plume, there is also a "banding" effect of cooler temperature (~1-2 K) and greatly enhanced moisture (>2 g/kg) at plume top. This banding effect is caused by a vertical displacement of the former continental boundary layer in the free troposphere in the FireOn simulation resulting from anomalous diabatic heating due to smoke absorption of sunlight that manifests primarily as a few hundred m per day reduction in large-scale subsidence over the ocean.





A large eddy simulation (LES) is then forced with free tropospheric fields taken from the outputs for the WRF-CAM5 FireOn and FireOff runs. Cases are run by selectively perturbing one variable (e.g., aerosol number concentration, temperature, moisture, vertical velocity) at a time to better understand the contributions from different indirect (microphysical), "large-

scale" semi-direct (above-cloud thermodynamic and subsidence changes), and "local" semi-direct (below-cloud smoke absorption) effects. Despite a more than five-fold increase in cloud droplet number concentration when including smoke aerosol concentrations, minimal differences in cloud fraction evolution are simulated by the LES when comparing the base case to a perturbed aerosol case with identical thermodynamic and dynamic forcings. A factor-of-two decrease in background free tropospheric aerosol concentrations from the FireOff simulation shifts the cloud evolution from a classical entrainment-

driven "deepening-warming" transition to trade cumulus to a precipitation-driven "drizzle-depletion" transition to open cells, however. The thermodynamic and dynamic changes caused by the WRF-simulated large-scale adjustments to smoke diabatic heating strongly influence cloud evolution in terms of both the rate of deepening (especially for changes in the inversion temperature jump and in subsidence) and in cloud fraction on the final day of the simulation (especially for the moisture "banding" effect). Such large-scale semi-direct effects would not have been possible to simulate using a small domain LES

model alone.

# 1 Introduction

## 1.1 Fundamentals of aerosol-radiation and aerosol-cloud interactions

Uncertainties relating to the interactions between airborne particles (aerosols) and clouds are the largest contributors to the overall uncertainty in quantifying the present-day radiative forcing due to human activities (Bellouin et al., 2020;

Sherwood et al., 2020). The "direct" radiative effect of aerosols is the result of their scattering or absorbing sunlight. Most aerosols are reflective and cool the planet by scattering sunlight back to space that otherwise would have been absorbed within the Earth system (Charlson et al., 1990). However, some aerosol species like black carbon and dust absorb a substantial amount of shortwave radiation as well (Bond et al., 2013; Haywood et al., 2004). Thus, whether the direct radiative effect of absorbing aerosol is positive (warming) or negative (cooling) is a function of both how absorbing the aerosol is as well as the albedo

(reflectivity) of the underlying surface (Chand et al., 2009; Chýlek and Coakley, 1974).

"Semi-direct" aerosol radiative effects are the result of rapid atmospheric thermodynamic adjustments to the direct effect and can be positive or negative depending on the relative distribution of the aerosol with respect to different types of clouds (Koch and Del Genio, 2010). Of greatest relevance to this work are the effects of absorbing aerosols either above or below shallow boundary layer clouds. Absorption below these clouds decreases relative humidity within the boundary layer,

reducing cloudiness (Ackerman et al., 2000; Hansen et al., 1997), whereas absorption above the clouds tends to strengthen the capping inversion, increasing cloudiness (Johnson et al., 2004). Semi-direct effects can be important for other cloud types as well: for instance, stabilization of the lower troposphere by heating aloft and shading of the surface can suppress convective cloud formation over land (Feingold et al., 2005; Jiang and Feingold, 2006; Sakaeda et al., 2011; Tosca et al., 2015) or,



alternatively, pronounced mid-level heating may destabilize the mid-to-upper troposphere and enhance convection over land

(Allen et al., 2019; Tummon et al., 2010). The radiative impact of direct effects alone is referred to as the radiative forcing due to aerosol-radiation interactions (ARI) and the combined impact of direct and semi-direct effects (rapid adjustments to the direct effect) as the effective radiative forcing due to ARI (Boucher et al., 2013).

Aerosol "indirect" effects refer to changes in radiation not from the aerosol optical properties themselves but rather from changes in cloud optical properties relating to the nucleation of liquid cloud droplets and/or ice particles. For liquid-phase

clouds (like subtropical marine stratocumulus), an increase in aerosol particles that can serve as cloud condensation nuclei (CCN) leads to an increase in the cloud droplet number concentration ($N_c$) under most conditions (Twomey, 1974). If the total amount of liquid water in the cloud remains the same, the effect of increasing the number of cloud droplets is to decrease their size, resulting in brighter (more reflective) clouds (Twomey, 1977). This phenomenon is essentially the result of maximizing the effective surface area of cloud droplets for a given volume of water and is often referred to as the first indirect effect or the

Twomey effect.

The microphysical cloud changes from the Twomey effect (greater number of smaller droplets) can lead to macrophysical cloud adjustments (changes in the total amount of cloud liquid and frequency of occurrence). Perhaps the most famous of these potential adjustments is the so-called lifetime effect, in which the shift in the cloud droplet size distribution toward smaller droplets decreases drizzle production (and thus the loss of cloud liquid), allowing clouds to last longer and

cover a greater areal extent (Albrecht, 1989; Christensen et al., 2020; Goren et al., 2019; Rosenfeld et al., 2019). This effect has also been labelled the second indirect effect, although this name is inappropriate in its implication that this effect is the only, or even the dominant, mechanism of cloud adjustments. Indeed, there exist several related adjustment mechanisms that oppose the effect of increased cloudiness by precipitation suppression, each involving in some form increases in the entrainment of warm, dry air that dissipates the cloud.

Entrainment rates can be enhanced via an evaporation effect because the phase relaxation timescale (timescale for evaporating droplets) decreases with increasing cloud droplet number (Jiang et al., 2006; Small et al., 2009; Wang et al., 2003), via a sedimentation effect in which the larger number of smaller (and thus more slowly settling) droplets increases the amount of water that can be evaporated in the entrainment zone (Ackerman et al., 2004; Ackerman et al., 2009; Bretherton et al., 2007), and via an increase in the maximum radiative cooling rate at cloud top (Williams and Igel, 2021). Entrainment may also be

enhanced by the suppression of precipitation itself — because drizzle tends to stabilize the marine boundary layer (MBL) via evaporative cooling, turbulence within the MBL and thus entrainment of free tropospheric air generally increases with decreasing drizzle (Wood, 2007). The vertical profile of sub-cloud evaporation matters, however, as drizzle that primarily evaporates just below cloud base can destabilize the sub-cloud layer and enhance convection (Feingold et al., 1996). The radiative impact of the Twomey effect alone is referred to as the radiative forcing due to aerosol-cloud interactions (ACI) and

the combined impact of the Twomey effect and adjustments as the effective radiative forcing due to ACI (Boucher et al., 2013).





## 1.2 Theories of the subtropical stratocumulus-to-cumulus transition

The nature and causes of the transition between overcast stratocumulus-dominated areas and lower cloud fraction cumulus-dominated areas of the subtropical oceans — the stratocumulus-to-cumulus transition (SCT) — has been a longstanding interest of the cloud physics community. Models describing, e.g., the well-mixed, stratocumulus-topped
boundary layer (Lilly, 1968) and the trade cumulus boundary layer (Albrecht et al., 1979) had been developed, but a fuller understanding of the processes behind the transition between these states remained elusive until the new observations and advances in numerical simulation of the 1990s. On the observational side, Lagrangian studies of deepening boundary layers off the Azores Islands as part of the Atlantic Stratocumulus Transition Experiment (ASTEX) provided compelling cases of a drizzling stratocumulus-topped MBL transitioning into a trade cumulus dominated layer and a more polluted, decoupled MBL
without the same degree of time evolution (Bretherton and Pincus, 1995; Bretherton et al., 1995). On the modeling side, two-dimensional cloud resolving models forced by increasing sea surface temperatures (SSTs) captured the evolution from a well-mixed layer with stratiform clouds to a transitional decoupled state with cumulus under stratocumulus to a trade cumulus layer (Krueger et al., 1995; Wyant et al., 1997). Fundamentally, these modeled SCTs were driven by enhanced entrainment induced by increasing latent heat fluxes with rising sea surface temperatures, creating more negative buoyancy fluxes below cloud base
and a weak stable layer (Bretherton and Wyant, 1997). Cumulus clouds detrain into the stratified upper layer and initially sustain the stratocumulus with moisture transport, but as SSTs continue to warm, the cumuli penetrate the inversion and mix in enough warm/dry free tropospheric air to dissipate the stratocumulus (Martin et al., 1995; Nicholls, 1984). It should be noted that these models did not allow for feedbacks between aerosols, cloud droplet concentrations, and precipitation.

Although earlier hypotheses about precipitation playing a necessary role in the SCT have not been borne out (Wang
et al., 1993), increasing drizzle in the stratocumulus clouds has been found to change the nature of the SCT by depleting the clouds of liquid water faster and inhibiting nocturnal recoupling of the MBL (Sandu and Stevens, 2011). Sandu and Stevens (2011) did not see a substantial change in the timing of the SCT due to drizzle in their simulations, however, as compared to those driven by initial lower tropospheric stability (LTS) changes. In contrast, more recent simulations with prognostic aerosol and cloud droplet concentrations that allow for collision-coalescence-induced aerosol-cloud-precipitation feedbacks show a
leading role for drizzle in driving rapid SCTs (Yamaguchi et al., 2017). With fully interactive aerosol, a positive feedback loop is able to develop in which greater precipitation scavenges aerosol, leading to clouds with lower numbers of larger droplets and thus greater precipitation, leading to yet greater aerosol depletion.

One important limitation of cloud resolving models and large eddy simulations (LES) is the difficulty in capturing cloud interactions with the large-scale environment and, depending on domain size, mesoscale circulations. To what extent
aerosol- and precipitation-related effects and feedbacks modify the "classical," entrainment-driven view of the SCT fundamentally driven by increasing SST with the timing set primarily by initial LTS remains an open and active area of research.



### 1.3 Smoke-cloud-climate interactions in the southeast Atlantic

The southeast Atlantic Ocean (SEA) is in many ways ideal for studying the coupled aerosol-cloud-climate system
because smoke from southern Africa's biomass burning season (roughly June-October) is advected over an area characterized
by a semi-permanent marine stratocumulus deck (Figure 1), opening up the possibility for strong and potentially competing
direct, semi-direct, and indirect aerosol radiative effects.

Human activities, particularly agricultural burning, are the major driver of global fire occurrence and trends (Andela
et al., 2017). Africa accounts for the vast majority of the world's burned area and approximately half of smoke carbon
emissions, with a slightly larger contribution from burning in subequatorial Africa than in northern Africa (Van Der Werf et
al., 2010). Many of the fires in southern Africa exhibit a clear diurnal cycle with a strong peak in the afternoon and almost no
activity at night (Roberts et al., 2009), which is consistent with what would be expected from controlled, agriculturally-driven
burning. Anthropogenic influence is even evident in the weekly cycle of burning — there is a pronounced decline in fire
activity on Sundays in Christian-dominated regions and a decline on Fridays in Muslim-dominated regions, especially in
croplands (Earl et al., 2015; Pereira et al., 2015).

As illustrated schematically in Figure 1, much of this smoke is transported away from the continent and overlies a
region of extensive stratocumulus cloud cover (Klein and Hartmann, 1993). Smoke aloft can lead to significant direct and
semi-direct aerosol effects. Satellite observations show that heating aloft coincides with thicker stratocumulus clouds over the
southeast Atlantic (Wilcox, 2010, 2012). Recent LES modeling work suggests that smoke absorption might need to occur very
close to the cloud tops to realize cloudiness increases through a stronger inversion, however (Herbert et al., 2020).

When smoke is mixed into the marine boundary layer, indirect aerosol effects and below-cloud semi-direct effects
can also become important. Evidence from recent observations at Ascension Island show lower cloud liquid water paths (LWP)
when more smoke is present within the MBL (Zhang and Zuidema, 2019). Some studies have shown decreases in cloud droplet
effective radius when smoke is present near clouds tops (Costantino and Bréon, 2010, 2013), although others have found a
more nuanced signal in terms of cloud and above-cloud smoke properties (Diamond et al., 2018; Painemal et al., 2014). There
are reasons not to expect strong correspondence between instantaneous above-cloud smoke properties and cloud properties, as
the characteristic timescale for mixing free troposphere (FT) aerosol into the MBL is on the order of days and cloud droplets
typically activate at cloud base, within the MBL (Bretherton et al., 1995; Diamond et al., 2018; Mardi et al., 2019).




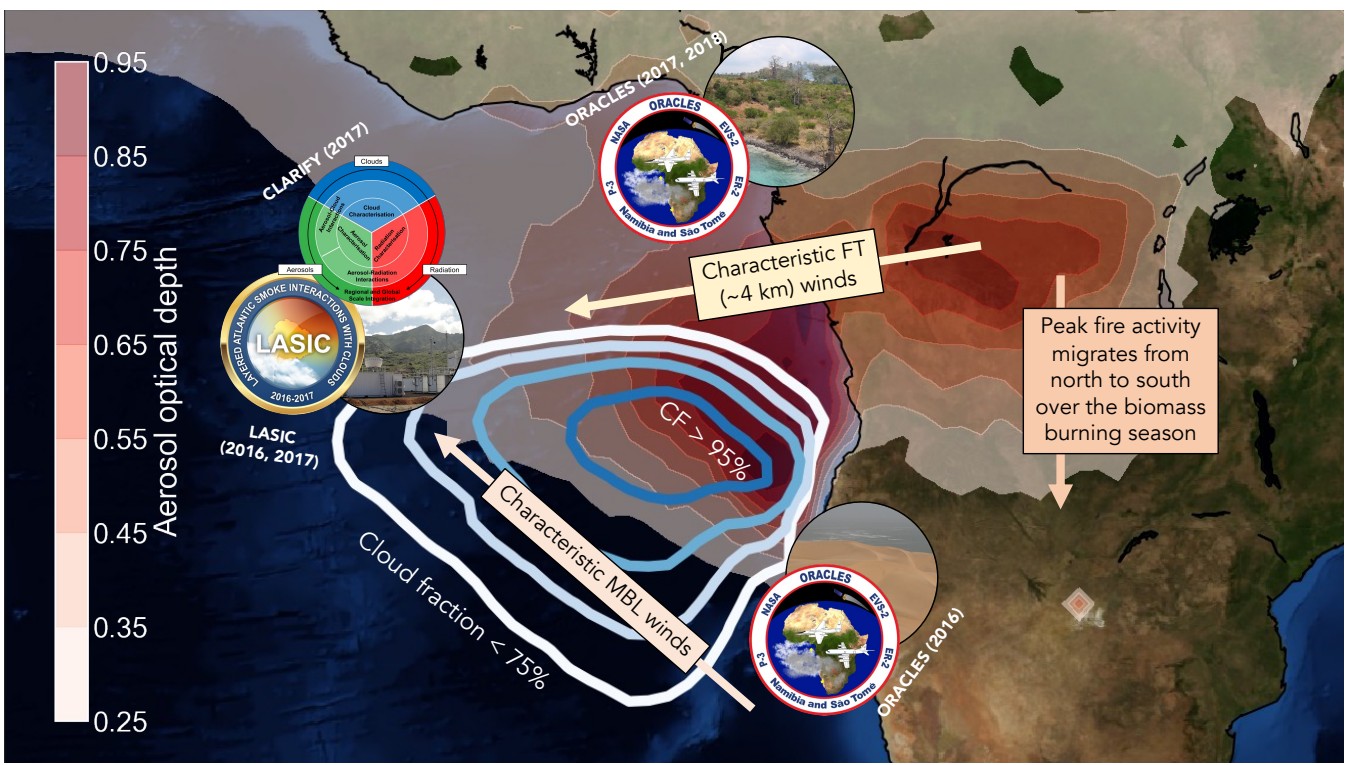

**Figure 1. Overview of the SEA smoke-cloud-climate system and recent campaigns. Smoke (shading; as represented by aerosol optical depth from the Moderate Resolution Imaging Spectroradiometer on Aqua) and cloud fraction (contours; from the Clouds and the Earth's Radiant Energy System Energy Balanced and Filled product) are averaged over the southern African biomass burning season (June-October) from 2003-2015. Logos and local images indicate the ORACLES deployment sites in Namibia (2016) and São Tomé (2017-2018). A logo for the CLARIFY campaign and logo and image from the LASIC deployment are placed near Ascension Island. Characteristic MBL winds, FT winds (at ~ 4 km), and seasonal fire migration are represented by schematic arrows. Background image is from the NASA Visible Earth Blue Marble collection.**

Complicating matters further, there is a strong seasonal cycle in burning and in the horizontal and vertical location of the smoke plumes over the southeast Atlantic — the plumes are located at progressively higher altitudes and are transported farther westward as the biomass burning season progresses and the southern African Easterly Jet strengthens (Adebiyi and Zuidema, 2016; Redemann et al., 2021; Zhang and Zuidema, 2021). Meteorological effects also complicate the analysis of smoke impacts because circulation patterns associated with smoke transport also influence cloud properties (Gaetani et al., 2021; Zhang and Zuidema, 2021) and because water vapor is enhanced within the smoke plumes as compared to other FT air (Adebiyi et al., 2015; Pistone et al., 2021), which can have separate and independent effects both aloft and when in contact with the clouds (Eastman and Wood, 2018).

As a result of this complexity, current global and regional climate models and large eddy simulation (LES) models disagree on everything from the magnitude and net sign of the direct radiative effect (Mallet et al., 2021; Zuidema et al., 2016) to the importance of semi-direct effects (Che et al., 2021; Ding et al., 2021; Gordon et al., 2018; Herbert et al., 2020; Sakaeda



et al., 2011) to the relative strengths of the first indirect effect and competing secondary indirect effects (Lu et al., 2018; Yamaguchi et al., 2015; Zhou et al., 2017). Most relevant for our analysis are two previous LES studies that have explicitly looked at the SCT in the context of an elevated smoke layer that subsides and is entrained into the boundary layer (Yamaguchi et al., 2015; Zhou et al., 2017). Absorbing aerosols are an interesting added complication for the SCT because they can influence inversion strength, MBL relative humidity and stability, cloud microphysics, and cloud macrophysics simultaneously and in competing manners. In the study of Yamaguchi et al. (2015), hereafter Y15, the SCT is delayed by smoke directly above the clouds because absorption in the free troposphere strengthens the inversion at cloud top and the smoke entrained into the boundary layer suppresses drizzle (and thus the positive feedback between precipitation and aerosol concentrations). Enhanced moisture in the smoke plume also helps sustain MBL clouds in the Y15 study by further reducing entrainment drying. In contrast, the SCT is sped up in the study of Zhou et al. (2017), hereafter Z17, because entrainment is enhanced by cloud top evaporative cooling with higher $N_c$. Aerosol concentrations were not coupled to precipitation sinks in Z17, thus precluding a drizzle-driven transition in that study. (Subsequent results with the model and setup used in Z17 show that including an aerosol loss term resulting from collision-coalescence can produce a rapid transition comparable to that of Y15, however; not shown). Interestingly, the presence of smoke causes net cooling in both studies despite the differences in the SCT speed, driven mostly by the delayed SCT in Y15 but by Twomey cloud brightening and longwave effects from reduced cloud fraction and a shallower MBL in Z17. Importantly, both Y15 and Z17 use background meteorology from the northeast Pacific (Sandu and Stevens, 2011) and thus do not include any effects of smoke-circulation interactions on scales larger than can be resolved within the ~10 km by 10 km LES domain, meaning potentially important characteristics of the southeast Atlantic's unique smoke-meteorology setup could not be captured in their results. Kazil et al. (2021) studied cases of transitions between closed-cellular to open-cellular convection using a reanalysis meteorological forcing from the southeast Atlantic, but their cases did not feature substantial smoke entrainment and thus did not address smoke indirect and below-cloud semi-direct effects.

A wealth of new observations and modeling studies (Zuidema et al., 2016) will be key to constraining the key physical processes and reaching a consensus on the net radiative effect of smoke over the southeast Atlantic during the biomass burning season. Earlier ground and aircraft campaigns — e.g., the Southern African Regional Science Initiative (SAFARI) campaigns in 1992 and 2000 —studied biomass burning effects in southern Africa (Fishman et al., 1996; Formenti et al., 2003; Garstang et al., 1996; Haywood et al., 2003; Hobbs, 2003; Sinha et al., 2004; Swap et al., 2003). However, these campaigns were restricted to the continent itself or else to a narrow region just off the coast except for two transit flights and one local flight around Ascension Island during the 1992 Transport and Atmospheric Chemistry near the Equator – Atlantic (TRACE-A) experiment (Fishman et al., 1996) and two transit flights during SAFARI-2000 (Haywood et al., 2003). Until recently, systematic observations of the remote southeast Atlantic Ocean during the biomass burning season have been mostly lacking.

Scientific interest and investment in the southeast Atlantic region have spiked in the past several years, featuring semi-coordinated American, British, French, German, Namibian, and South African aircraft and ground campaigns (Zuidema et al., 2016). Here we focus on three that sampled the remote southeast Atlantic during August 2017. The NASA ObseRvations of Aerosols above CLouds and their intEractionS (ORACLES) campaign brought a P3 Orion turboprop aircraft instrumented





with a variety of aerosol and cloud probes and remote sensors to the SEA in September 2016, August 2017, and October 2018, the first based out of Walvis Bay, Namibia, and both later deployments out of São Tomé, São Tomé e Príncipe (Redemann et al., 2021). A consortium of several British universities, the UK Met Office, and other partners flew the Facility for Airborne Atmospheric Measurements (FAAM) BAe-146 aircraft out of Ascension Island (7.9° S, 14.4° W) during the CLouds and Aerosol Radiative Impacts and Forcing: Year 2017 (CLARIFY) campaign in August-September 2017 (Haywood et al., 2020).

The US Department of Energy Atmospheric Radiation Measurement (ARM) Mobile Facility 1 was also deployed to Ascension Island for the Layered Atlantic Smoke Interactions with Clouds (LASIC) campaign from 1 June 2016 to 31 October 2017 (Zuidema et al., 2018a; Zuidema et al., 2018b). A schematic of the location of each deployment is included in Figure 1.

In 2017 the NASA P3 was mainly deployed from São Tomé, but a joint flight between the P3 (flight PRF05Y17) and the FAAM BAe (flight C031) out of Ascension Island took place on 18 August 2017 (Figure 2). The flight observations show

a moderately polluted MBL but relatively clean conditions aloft (Barrett et al., 2022), suggesting the MBL smoke must have been entrained earlier as the MBL air transited the SEA. As discussed below, smoke observed further south and east on an ORACLES flight on 15 August 2017 may have contributed to the MBL smoke population sampled on 18 August 2017. Combined with ground measurements from the ARM facility, these synergistic ORACLES-CLARIFY-LASIC observations offer a great opportunity for a modeling case study with abundant observational constraints.

The complex interplay of smoke effects on the transition between overcast stratocumulus and broken cumulus cloud fields during this well-observed period in mid-August 2017 is the main subject of our investigation. The rest of this manuscript is organized as follows. Section 2 describes our general approach, definitions of key variables analyzed, observational data used, and details of the model setups. In order to capture the effects of biomass burning from the synoptic to local scales, we first run a regional climate model and then force a large eddy simulation with the regional model output. Section 3 describes

the regional climate results and Section 4 the LES results. Section 5 compares the base cases of the regional and LES output to observations from the ORACLES-CLARIFY joint flight, LASIC ground site, and geostationary satellite retrievals. Section 6 provides further discussion and a summary of our main conclusions.



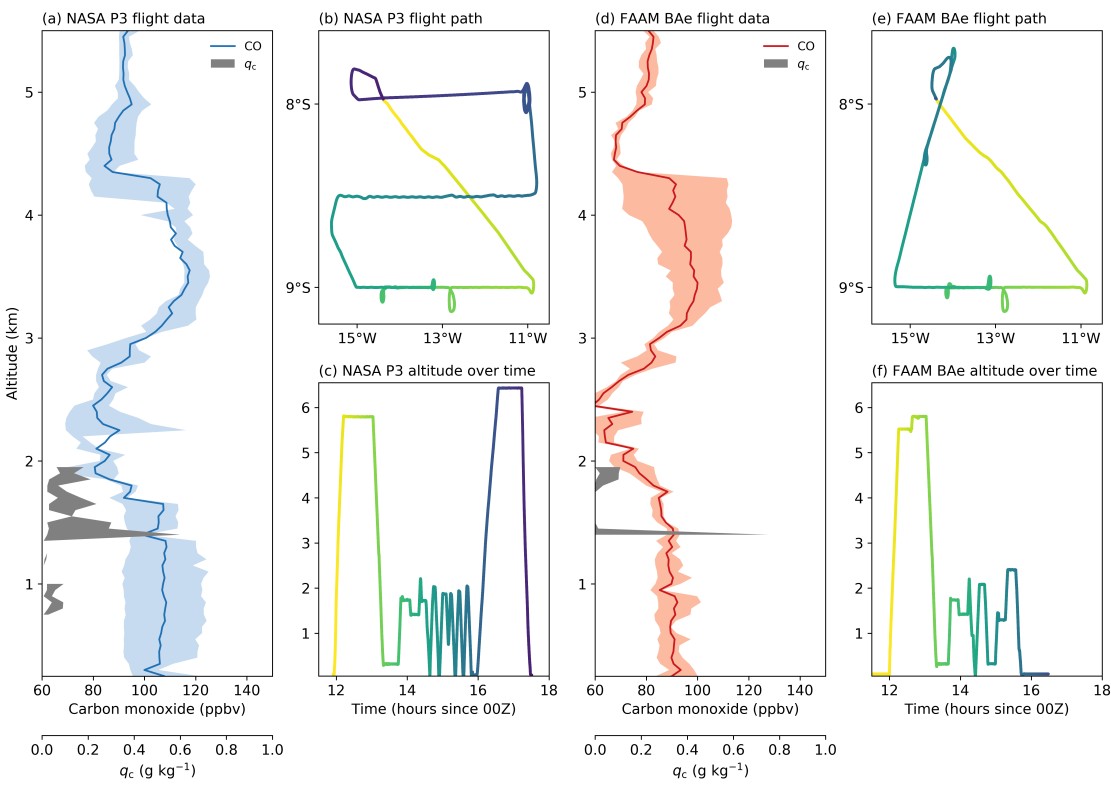

**Figure 2.** Summary of the 18 August 2017 ORACLES-CLARIFY joint flight. (a) Mean (line) and interquartile range (shading) of carbon monoxide and interquartile range (grey shading) of cloud water content observed over the course of the NASA P3 flight. Flight tracks are displayed in terms of (b) latitude and longitude and (c) time (color) and altitude for the NASA P3. (d-f) As in (a-c), but for the FAAM BAe.

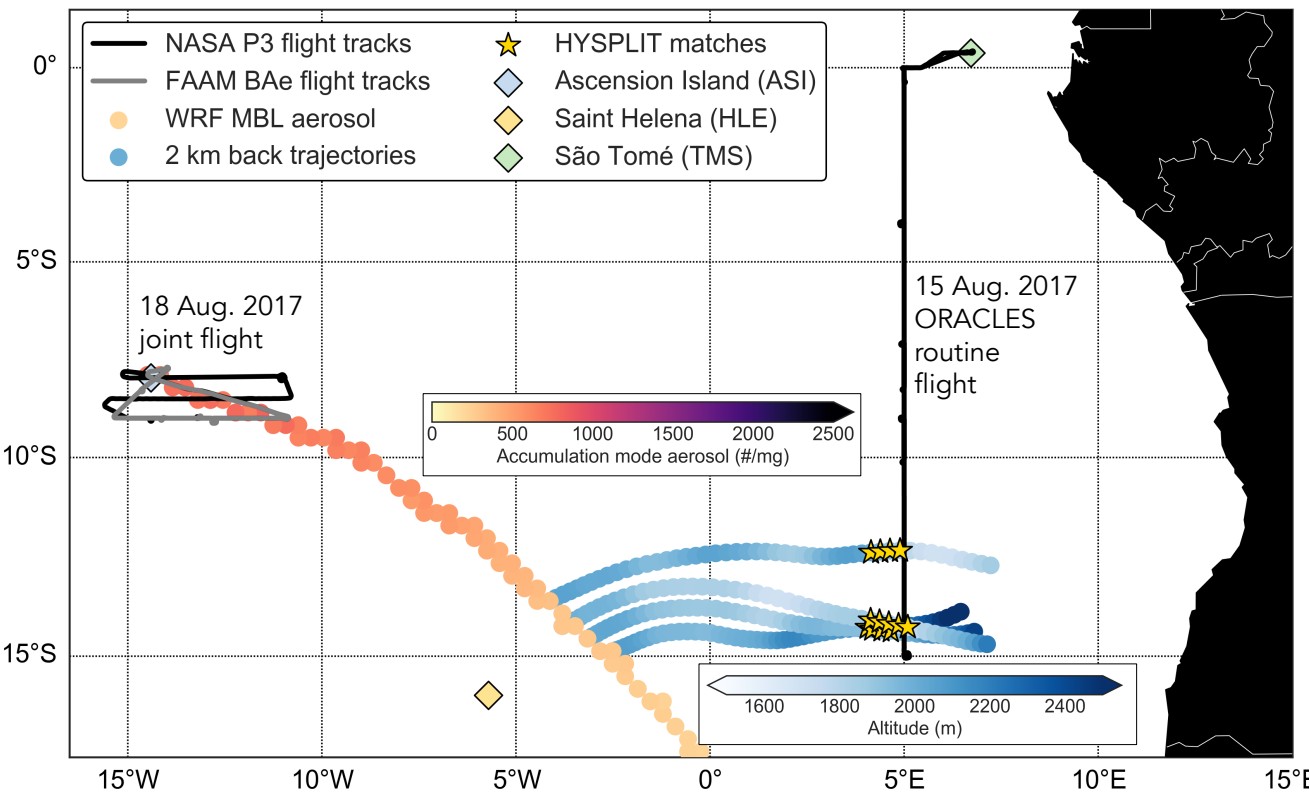

**Figure 3. Case study overview.** Map of the Lagrangian MBL trajectory (orange points) ending at Ascension Island along with the flight tracks from the NASA P3 (black lines) on 15 and 18 August 2017 and the FAAM BAe (gray line) on 18 August 2017. HYSPLIT back trajectories initialized at 2 km that intersected with the 15 August 2017 ORACLES flight at an appropriate location and time are colored in blue (tint corresponds to altitude) and the matches are indicated by stars. Shading along the trajectory from peach to red-orange indicates the MBL smoke concentration in the WRF-CAM5 FireOn case. Locations of Ascension Island, Saint Helena, and São Tomé are indicated by markers.

## 2 Methods

### 2.1 General approach

To capture the evolution of the cloud field observed during the joint flight from its presumed starting point in the overcast stratocumulus region, a Hybrid Single Particle Lagrangian Integrated Trajectory Model (HYSPLIT) back trajectory is run isobarically for three days using regional model winds (more details of the regional model are provided below) after being initialized at 500 m at the location of Ascension Island at 20:00 UTC (Stein et al., 2015). Figure 3 shows the path of this Lagrangian (air-mass-following) trajectory.

An ORACLES flight sampling at a near-constant longitude of 5° E occurred on 15 August 2017 (PRF03Y17). HYSPLIT back trajectories using the National Center for Environmental Prediction's Global Data Assimilation System 0.5° x 0.5° meteorology were initialized every hour along the Lagrangian trajectory at 2 km and run for 48 hours to test if any of



the smoke plumes that could be entraining into the MBL along the SCT trajectory of interest may have been sampled on the earlier flight. Matches were considered close enough to contain useful information if they intersected the ORACLES flight track within a degree of longitude on either side of 5° E from 07:00 to 18:00 UTC on 15 August 2017. Several matches (stars in Figure 3) occurred between 12-15° S from 08:00 to 12:00 UTC. Thus, smoke properties on the 15 August 2017 flight can serve as a loose constraint on the properties of smoke that had been previously entrained into the MBL sampled on 18 August 2017.

A regional climate model is first used to simulate the synoptic scale interactions between smoke from the continent and clouds over the ocean and then a large eddy simulation model is forced with regional model output and used to explore cloud transitions under the influence of smoke in greater detail. The regional model resolution is too coarse to resolve boundary layer clouds and their dynamics, necessitating the use of uncertain parameterizations, making its cloud results suspect, whereas the LES domain is too small to resolve smoke-induced changes in the large-scale circulation that can greatly affect the SCT. By providing the LES with large-scale forcing from the regional climate model, we are able to take advantage of the primary advantages of each.

## 2.2 Campaign data

Instruments used in this work and any data processing are reviewed briefly below. Full lists of instrumentation are provided in Appendix B of Redemann et al. (2021) for the NASA P3 aircraft during ORACLES, in Haywood et al. (2020) for the FAAM BAe aircraft during CLARIFY, and in Zuidema et al. (2018b) for the AMF1 site during LASIC. All measurements originally reported in terms of volume mixing ratios are converted to mass mixing ratios (which are conserved for adiabatic processes) using either the ambient air density or the density of air at standard temperature and pressure (1.29 kg m$^{-3}$), as appropriate.

Total water mixing ratio ($q_T$) and liquid water potential temperature ($\theta_l$; the temperature that would result after complete evaporation of all liquid water in an air parcel brought adiabatically to 1000 hPa) are the primary thermodynamic variables analyzed. Liquid water potential temperature and total water mixing ratio are useful variables for studying MBL motions as they are conserved in moist adiabatic processes (Betts, 1973; Jones et al., 2011).

Specific humidity ($q_v$) measurements for ORACLES are from the Picarro Incorporated isotopic water vapor analyzer (Gupta et al., 2009). Similar measurements are available from the standard NASA P3 aircraft instrumentation package as well as from an ABB/Los Gatos Research CO/CO2/H2O Analyzer (Pistone et al., 2021), but the Picarro instrument had the fewest data outages on the 18 August 2017 flight. Cloud water mixing ratio ($q_c$) measurements are from the King hot wire probe (King et al., 1978). The total water mixing ratio is calculated as the sum of the Picarro water vapor and King liquid water. For CLARIFY, specific humidity is from a SpectraSensors Water Vapor Sensing System (WVSS-II) and total condensed water content with a Nevzorov hot wire probe (Korolev et al., 1998). Liquid water potential temperature is approximated using the static ambient temperature and the cloud liquid water mixing ratio as:





$$\theta_1 \approx \theta - \frac{L_v}{c_p} q_c, \tag{1}$$

where $\theta$ is the potential temperature, $L_v$ is the latent heat of vaporization and $c_p$ is the specific heat of dry air at constant pressure.

The inversion height ($z_i$) is defined as the altitude below 3 km with the greatest vertical gradient in liquid water potential temperature. Inversion jumps are calculated by taking the difference of the average values of the quantity of interest 100 to 200 m above and below the inversion. The decoupling parameter used, $\Delta q_T$, is defined as the difference between the
total water mixing ratio from the bottom and top 25% of the MBL (Jones et al., 2011).

Cloud droplet number concentration ($N_c$) for droplets with radii of 3-50 μm was measured using a Droplet Measurement Technologies (DMT) Cloud Droplet Probe (CDP) for both ORACLES and CLARIFY. For ORACLES, aerosol size distributions (subset for particles between 60 and 600 nm in diameter) were measured with a DMT Ultra-High Sensitivity Aerosol Spectrometer (UHSAS) and empirically corrected for an undersizing issue related to the presence of refractory black
carbon (Howell et al., 2021). For CLARIFY, aerosol number concentration ($N_a$) was measured with a DMT Passive Cavity Aerosol Spectrometer Probe (PCASP) for particles between 0.1 and 3 μm in diameter. For LASIC, $N_a$ (subset for particles between 60 and 600 nm in diameter and empirically corrected for undersizing) was calculated using a UHSAS (Arm User Facility, 2016b) from the AMF1 site at 340 m elevation.

Refractory black carbon (rBC) from 53 to 524 nm mass equivalent diameter was measured using a DMT single-
particle soot photometer (SP2) with a solid diffuser inlet (Arm User Facility, 2016a; Schwarz et al., 2006; Stephens et al., 2003) for all three campaigns. For ORACLES, single-scatter albedo ($\omega_0$) at 550 nm is calculated with scattering data from a TSI Incorporated integrating nephelometer and absorption data from a Radiance Research Particle Soot Absorption Photometer (Pistone et al., 2019) and organic aerosol mass concentration was measured by a High-resolution Aerodyne Aerosol Mass Spectrometer operating in V-mode (Canagaratna et al., 2007; Dobracki et al., 2022).

Vaisala RS-92 radiosondes were launched from the Ascension Island airport (near sea level) at 00:00, 06:00, 12:00, and 18:00 UTC each day during the LASIC campaign in August 2017 (Arm User Facility, 2016c). The radiosondes were swiftly swept away from the island by the prevailing winds (Zhang and Zuidema, 2019).

In addition to the aircraft and ground measurements, cloud properties ($N_c$, cloud phase, and LWP) from the Spinning Enhanced Visible and Infrared Imager (SEVIRI) aboard the geostationary Meteosat-10 satellite were retrieved by the NASA
Langley Research Center (Painemal et al., 2012).

## 2.3 Model setups

### 2.3.1 Regional climate modeling: WRF-CAM5

For our regional climate model, we use the Weather Research and Forecasting model (WRF) coupled with chemistry (Chem )configured with the Community Atmosphere Model version 5 (CAM5) physics package (Ma et al., 2014). WRF-
Chem-CAM5 (hereafter shortened to WRF-CAM5 or just WRF) was run at 36 km x 36 km horizontal resolution with 74





vertical levels over a domain spanning from approximately 40° S to 15° N and 30° W to 50° E. Separate parameterization schemes are used for deep convection (Zhang and Mcfarlane, 1995) and shallow cumulus convection and turbulence (Park and Bretherton, 2009), although the shallow convection scheme is turned off in these simulations as it was found to lead to strong suppression of cloud formation and MBL growth. One two-moment microphysics scheme (Gettelman and Morrison, 2008)

and macrophysics scheme (Park et al., 2014) are used for shallow clouds and another two-moment scheme for convective clouds (Lim et al., 2014; Song and Zhang, 2011; Zhang and Mcfarlane, 1995). Effects of convective entrainment on aerosol activation are included for deep convective clouds (Barahona and Nenes, 2007). A modal aerosol module (Liu et al., 2012) with Aitken, accumulation, and coarse modes (MAM3) coupled with a gas phase chemistry scheme (Zaveri and Peters, 1999) is used for smoke (and other aerosol) properties. Aerosol particles are assumed to be internally mixed and optical properties

are calculated by dividing the MAM3 aerosol modes into eight size bins and applying Mie theory calculations (Fast et al., 2006). The cloud droplet activation scheme accounts for activation of giant CCN and insoluble particles such as dust and black carbon (Fountoukis and Nenes, 2005; Zhang et al., 2015). Smoke emissions are from the Quick Fire Emission Dataset (QFED), version 2 (Darmenov and Da Silva, 2015) with a representative diurnal cycle of burning applied and include black carbon, organic carbon, sulfur dioxide, ammonia, and nitrogen oxides among other relevant species. Interpolation and the calculation

of some diagnostic variables (such as low, middle, and high cloud fraction) were performed via the wrf-python software package produced by the University Corporation for Atmospheric Research/National Center for Atmospheric Research (Ladwig, 2017).

The same configuration of WRF-CAM5 used here was evaluated in two observation-model intercomparison studies (Doherty et al., 2022; Shinozuka et al., 2020) for monthly climatologies using ORACLES measurements, and further details

on the configuration can be found there. The evaluation showed that WRF-CAM5 tends to rank within the best of the evaluated models for variables like smoke concentrations, optical properties, and plume location. An exception is that WRF-CAM5 tends to overestimate aerosol size (and thus underpredict Ångström exponent). Thus, the size distribution in this work is prescribed from observations (see next sub-section). These studies also show that WRF-CAM5 ranks within the best models for cloud coverage, optical thickness and top heights.

WRF-CAM5 was run for three cases, summarized in





Table 1:

1. FireOn, in which estimated biomass burning emissions from south-central Africa are included and can interact with the radiation and cloud microphysical schemes;

2. FireOff, in which no biomass burning emissions are included in south-central Africa;

3. RadOff, in which biomass burning emissions are included and can interact with the cloud microphysical scheme but are excluded from the radiation scheme.





**Table 1. Differences between the WRF-CAM5 FireOn, FireOff, and RadOff cases. X marks indicate that a phenomenon is included**
**in the simulation, whereas the lack of a mark indicates that the phenomenon is excluded.**

| Case name | QFED fire emissions? | Aerosol-radiation interactions? | Smoke-radiation interactions? | Aerosol-cloud microphysical interactions? | Smoke-cloud microphysical interactions? |
|---|---|---|---|---|---|
| FireOn | X | X | X | X | X |
| FireOff |  | X |  | X |  |
| RadOff | X |  |  | X | X |

The FireOn and FireOff models are initialized identically at 00:00 UTC on 15 July 2017 from 1º x 1º resolution National Center for Environmental Prediction (NCEP) Final Operational Global Analysis meteorology and Copernicus Atmosphere Monitoring Service (CAMS) reanalysis aerosol. Boundary conditions are set by the NCEP meteorological

reanalysis and CAMS aerosol as well. Every five days, the models are reinitialized with NCEP meteorology; chemical and aerosol properties are not reinitialized (Doherty et al., 2022; Shinozuka et al., 2020). Thus, any smoke initially present in the FireOff simulation would have been transported outside the region by the time period of interest, although any long-range transport of biomass burning aerosol from regions other than southern Africa captured in the CAMS boundary conditions would be included. The output analyzed in this chapter begins on the 00:00 UTC, 14 August 2017, reinitialization and includes

free running output until 00:00 UTC on 21 August 2017. We do not use output from the reinitialization on 19 August 2017.

For the RadOff case, we start with the reinitialization of FireOn at 00:00 UTC on 14 August 2017 and then run the model freely until 00:00 UTC on 21 August 2017, but without any aerosol-radiation interactions. Any effects from aerosol-radiation interactions prior to 14 August 2017 are thus identical between the cases, meaning all differences between the FireOn and RadOff simulations are due to processes occurring from 14-21 August 2017.

The Lagrangian trajectory over which the stratocumulus-to-cumulus transition is to be studied is identified by running an isobaric back trajectory initialized at 500 m over Ascension Island at 20:00 UTC on 18 August 2017 using HYSPLIT on the WRF FireOn horizontal winds for three days. Curtains are compiled over the course of the trajectory (averaging over a 3º x 3º region centered at the trajectory location) and linearly interpolated to hourly resolution. FireOff and RadOff curtains are created using the trajectory locations from FireOn.

Forcing files for the large eddy simulation are created from the FireOn and FireOff curtains. (As will be discussed later, FireOff and RadOff free tropospheric thermodynamic and dynamic properties are essentially equivalent.) Forcing files include time-altitude profiles of absolute temperature, total water mixing ratio, horizontal and geostrophic winds, vertical and pressure velocity, accumulation mode aerosol mass and number concentration, and ozone mixing ratio and time series of surface pressure and sea surface temperature. WRF-CAM5 output is interpolated onto a vertical grid with 5 hPa resolution

from 1010 hPa to 550 hPa. When clouds are present, the temperature and specific humidity fields are converted to liquid water temperature and total water mixing ratio. To reduce noise, the winds used for nudging are calculated by first taking the time-





average (00:00 UTC 14 August 2017 to 00:00 UTC 21 August 2017) wind at each grid point and then creating a curtain based only on the location of the trajectory. Vertical winds are further smoothed by imposing the trajectory-mean vertical profile at every time step. This smoothing is necessary to avoid results being overly influenced by high-frequency perturbations that are

likely noise (Diamond, 2020). Temperature and moisture advection are neglected as the FT evolution over the (MBL) Lagrangian trajectory is fully captured in the curtains and a relatively rapid nudging timescale (three hours) is employed in the large eddy simulations.

If the MBL evolution of temperature and moisture, in particular, were left unchanged in the LES forcing file, it would be essentially impossible for the boundary layer to fall below the WRF forcing value in the LES, because if that situation were

to occur, the MBL would begin entraining "FT" air nudged to have MBL-like properties. For example, if WRF simulated an MBL height of 1500 m but SAM would only produce an MBL height of 1200 m, the nudging scheme would impose WRF MBL values in the SAM FT. Therefore, after the first (initialization) timestep, MBL values of (liquid water) temperature, total water mixing ratio, accumulation mode aerosol number and mass, and ozone in the LES forcing are replaced by FT values that have been linearly extrapolated to 1010 hPa. For the smoke parameters, extrapolated values are restricted to be between the

maximum FT value near the MBL top and a "background" minimum to avoid unphysical values. The LES is only nudged in the FT, so forcing values within the LES-diagnosed MBL do not have any effect on the simulations.

Above 550 hPa, a temporally constant vertical profile of all dynamic, thermodynamic, and chemical variables from the August 2017 average calculated from the Modern-Era Retrospective analysis for Research and Applications, Version 2 (MERRA-2) reanalysis (Gelaro et al., 2017), averaged over a region from 0-20º S and 15º W to 5º E, is used to calculate

downwelling shortwave and longwave radiative fluxes. This uniform profile is applied to all forcings to ensure that differences in the evolution of the LES cloud fields can be attributed solely to the processes occurring within the lower troposphere that are of interest in this study.

### 2.3.2 Large eddy simulation: SAM

The System for Atmospheric Modeling (SAM) is a cloud resolving model/large eddy simulation model (depending

on horizontal resolution) with an anelastic dynamical core on an Arakawa C-type grid that has been widely used in the ACI and SCT literature (Khairoutdinov and Randall, 2003). We follow a recently-developed approach (Goren et al., 2019; Kazil et al., 2021; Narenpitak et al., 2021) in which Lagrangian LES are forced by the large-scale meteorology from a reanalysis product or regional or climate model, WRF-CAM5 in the present case. We use a uniform horizontal resolution of 50 m and a stretched vertical grid of 324 grid points from the surface to 7 km with resolution varying from 5-20 m below 4 km to 100-

300 m above 4 km. Simulations are performed on a domain of 240 grid points in each horizontal direction (12 km x 12 km). The standard time step is set to 1 second but is adaptively shortened to avoid numerical instability when necessary. The first four hours of the simulation (20:00 UTC on 15 August 2017 to 00:00 UTC on 16 August 2017) are used to spin up the case and are excluded from the analysis.



Radiation is calculated every 12 seconds along the trajectory, corresponding to a spatial distance of approximately

100 m, with the rapid radiative transfer model (Mlawer et al., 1997) modified to use the simulated properties below 7 km and

a constant atmospheric profile above. The solar zenith angle varies diurnally and with the latitude and longitude of the

Lagrangian trajectory. Sub-grid turbulence closure is from a prognostic 1.5 order turbulent kinetic energy (TKE) scheme

(Deardorff, 1980). Turbulent surface fluxes are computed prognostically following Monin–Obukhov similarity theory (Foken,

2006; Monin and Obukhov, 1954). Advection of scalars is performed with a monotonic multidimensional fifth-order

conservative advection scheme (Yamaguchi et al., 2011) and momentum with a third-order Adams-Bashforth time integration

method (Durran, 1991) coupled with the second-order centered advection scheme.

For aerosol and cloud microphysics, we use a bin-emulating bulk two-moment scheme (Feingold et al., 1998; Wang

and Feingold, 2009), as in Yamaguchi et al. (2017). Smoke-radiation interactions are represented with a lookup-table method

as in Y15. A single log-normal accumulation mode meant to represent bulk smoke properties with a source from free-

tropospheric entrainment is created using representative values from the ORACLES-2017 deployment (Fig. 4). From the

ORACLES-2017 UHSAS data within smoke plumes (defined as data with organic aerosol mass concentrations exceeding 10

$\mu g \; kg^{-1}$), the typical geometric mean diameter ($D_g$) of the aerosol number distribution is estimated as 185 nm and the geometric

standard deviation ($\sigma_g$) as 1.5. The hygroscopicity parameter is specified as 0.2, which is lower than that reported from the

ORACLES-2016 deployment (Kacarab et al., 2020), but is consistent with adjustments to the previously calculated values

given the UHSAS undersizing issues (Howell et al., 2021) and to values for primarily carbonaceous aerosol used in other

models (Fanourgakis et al., 2019). The modal (dry) single-scatter albedo is 0.85 at 550 nm, which we adopt for most of the

smoke-including simulations. The average August 2017 single-scatter albedo was measured as a more absorbing 0.8 at

Ascension Island during the LASIC campaign, however (Zuidema et al., 2018a). Absorption and scattering measurements

were more comparable between the NASA P3 and FAAM BAe aircraft than between either aircraft and the ground

measurements during the joint flight, which could reflect instrument differences or differences in the aerosol population aloft

and near the surface (Barrett et al., 2022). Accumulation mode aerosol number is specified from the WRF-CAM5 forcing files.

We also include a wind-dependent surface aerosol source (Clarke et al., 2006; Kazil et al., 2011) that is assumed to have the

same size distribution and hygroscopicity as the FT aerosol, as in Y15. (Note that although both parameters should be different

for sea salt in reality, for our purposes the surface source aerosol and smoke aerosol are identical.)




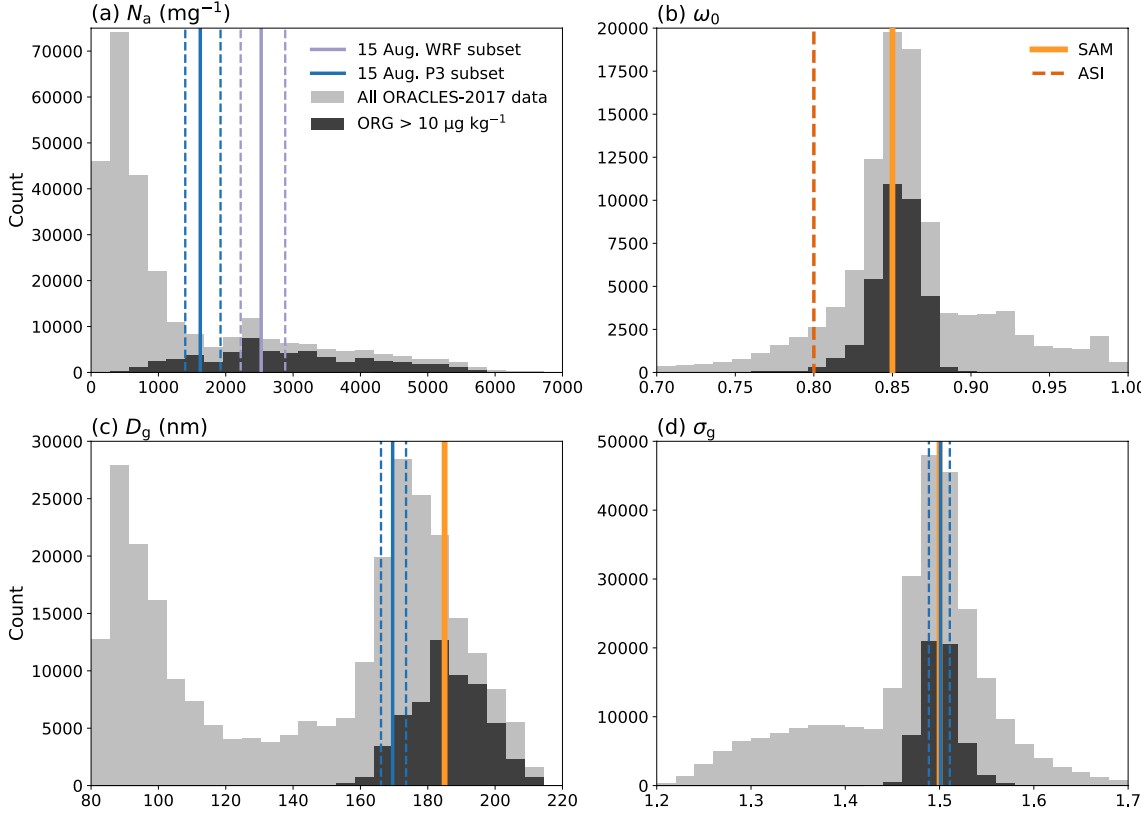

**Figure 4. ORACLES-2017 flight statistics and assumed aerosol size distribution and optical properties for SAM. All flight data is represented as light grey histograms and data with organic aerosol concentrations (ORG) greater than 10 μg kg⁻¹ as dark grey**
**histograms for (a) aerosol number concentration, (b) dry single-scatter albedo at 550 nm, and the (c) geometric mean diameter and (d) geometric standard deviation of the aerosol size distribution. Blue solid and dashed lines denote mean and interquartile values, respectively, for the 15 August 2017 flight between 1100 and 1300 UTC for ORG > 10 μg kg⁻¹. The purple solid and dashed lines in (a) denote the mean and interquartile values, respectively, of WRF-CAM5 FireOn aerosol number concentration subset along the P3 flight track from 1100 to 1300 UTC on 15 August 2017 for ORG > 10 μg kg⁻¹. Values selected for use in the standard SAM aerosol**
**representation are shown as gold lines. The dashed orange line in (b) represents the average single-scatter albedo value observed during the LASIC campaign (ASI) for August 2017.**

As can be seen in Fig. 4a, WRF-CAM5 simulated aerosol number concentrations about 50% larger than those observed during the 15 August 2017 P3 flight between 1100 and 1300 UTC (the most relevant times based on the HYSPLIT

trajectories and flight path). Figure S1 shows WRF-CAM5 FireOn curtains at the P3 time/location along the flight path for $N_a$ as well as black and organic carbon mass concentration, revealing that the overestimate in model number is not necessarily due to an overestimate of smoke mass in general (indeed, the model appears to underestimate smoke mass for part of the plume). Thus, WRF-CAM5 may have a high bias in $N_a$, at least for the time period simulated, meaning the standard FT $N_a$ forcings for the LES may represent an unrealistically large supply for this case. The WRF-CAM5 FireOn $N_a$ values are well

within the standard range of in-plume observations during ORACLES-2017 in general, however.



The free troposphere is nudged to the forcing temperature, moisture, and aerosol properties on a three-hourly timescale. Nudging begins 100 m above the maximum vertical gradient in liquid water static energy (equivalent to a definition via maximum absolute $\theta_l$ gradient) and ramps up over a 200 m buffer region (Blossey et al., 2013). MBL properties are not nudged and evolve freely. Horizontal winds are forced with the geostrophic wind fields and are nudged to the model winds in the FT on a timescale of six hours. Because we strongly nudge the FT temperature, aerosol-radiation interactions simulated by SAM primarily influence the MBL. Maintaining the heating of the FT by aerosol within SAM would amount to "double-counting" of the aerosol heating effect, as it is already accounted for in the FireOn forcing.

Runs forced exclusively with output from FireOn or FireOff are referred to as AllOn and AllOff, respectively. To test the effects of individual variables, runs are also performed with modified forcings based on FireOn but with one variable of interest taken from FireOff. As an example, the "TOff" run is forced with FireOn output for all variables except temperature, which is instead taken from FireOff.





Table 2 summarizes each of these cases, as well as two other sensitivity tests: "AllOff$_{N/2}$", in which the background aerosol number concentration in the FireOff forcing is reduced by a factor of two; and "AsiAbs", in which the single-scatter albedo is set at the LASIC-measured value of 0.8 (Zuidema et al., 2018a). Aerosol-radiation interactions in the MBL are represented using the lookup tables for the AllOn family of runs and are excluded in AllOff and AllOff$_{N/2}$. Including aerosol-radiation interactions in AllOff using smoke optical properties from AllOn results in negligible changes to cloud evolution (not shown).

Our 12 km x 12 km x 7 km domain with 50 m horizontal grid spacing is similar in size to the 12 km x 12 km x 4.25 km domain with 50 m horizontal grid spacing used in Y15 and the 10.8 km x 10.8 km x 3.2 km domain with 75 m horizontal grid spacing used in Z17. As in Y15, but unlike Z17, our setup allows for wet scavenging of aerosol and thus for aerosol-precipitation interactions. Both Y15 and Z17 were forced by the Sandu & Stevens (2011) trajectory based on typical SCT conditions from the northeast Pacific. The work presented here represents an advance by using forcings derived specifically for the SEA region. Unlike the SEA-specific forcings from Kazil et al. (2021), our case also features substantial smoke entrainment into the MBL and thus can address questions of smoke indirect and below-cloud semi-direct effects. Our forcings derived from the WRF FireOff case also allow us to isolate smoke effects in a manner that cannot easily be done with reanalysis data.





**Table 2. Source (WRF-CAM5 FireOn or FireOff) of the variable in the forcing and value of the single-scatter albedo for each SAM case. The variable perturbed from the AllOn or AllOff case is italicized.**

| Case name | $N_a$ | $T_l$ | $q_T$ | $w$ | $\omega_0$ |
|---|---|---|---|---|---|
| **AllOn** | FireOn | FireOn | FireOn | FireOn | 0.85 |
| **AllOff** | FireOff | FireOff | FireOff | FireOff | - |
| **AllOff$_{N/2}$** | *(FireOff)/2* | FireOff | FireOff | FireOff | - |
| **NOff** | *FireOff* | FireOn | FireOn | FireOn | 0.85 |
| **TOff** | FireOn | *FireOff* | FireOn | FireOn | 0.85 |
| **QOff** | FireOn | FireOn | *FireOff* | FireOn | 0.85 |
| **WOff** | FireOn | FireOn | FireOn | *FireOff* | 0.85 |
| **AsiAbs** | FireOn | FireOn | FireOn | FireOn | *0.8* |


## 3 The stratocumulus-to-cumulus transition modified by smoke in the WRF-CAM5 regional climate model

### 3.1 Lagrangian perspective for the 15-18 August 2017 case study

Figure 5 shows Lagrangian curtains of liquid water potential temperature, total water mixing ratio, and accumulation mode aerosol number concentration (defined here as including activated particles in cloud droplets) following the MBL
trajectory ending at 500 m altitude at Ascension Island on 20:00 UTC, 18 August 2017 [day-of-year (doy) 230], for the WRF-CAM5 FireOn case (first row). In the model, an elevated smoke plume is not yet in contact with the MBL at the start of the study period [21:00 UTC on 15 August 2017 (doy 227)]. The sharp jumps apparent in the $\theta_l$ and $q_T$ fields demarcate the inversion layer separating free tropospheric and marine boundary layer air. Contact between the MBL and the denser portion of the plume (indicated by an aerosol number concentration of 1000 mg$^{-1}$) is established during the second night and a thin
sliver of smoke remains above the MBL by the end of the study period. A relatively weak smoke plume also emerges aloft during the final day but is too high in altitude to interact with the MBL directly. Although the amount of smoke directly above the MBL peaks during the second day, the smoke concentration within the MBL increases steadily over the course of the Lagrangian trajectory. This is because the timescale for entraining free tropospheric air into the MBL is generally on the order of days, so MBL aerosol properties at any given point in time represent the time integral of what has been entrained (and lost
to wet and dry deposition, emitted from the sea surface, etc.) over the MBL history (Diamond et al., 2018).

The differences between the curtains of $\theta_l$, $q_T$, and $N_a$ between the FireOn and FireOff are shown in the second row of Figure 5 (absolute values for the FireOff trajectory are provided in Figure S2d-f). As would be expected from smoke absorption of sunlight, the core of the smoke plume in FireOn is warmer than the FT in FireOff by several Kelvin. However, there is also a "banding" effect of ~1-2 K cooler temperatures at the plume top which is not easily explained by local smoke-
radiation interactions. A similar feature was simulated by the UK Met Office Unified Model in another case study of the



southeast Atlantic, but it remained well above the MBL in that case (Gordon et al., 2018). There appears to be some (<1 g/kg) enhancement of water vapor within the core of the plume, although the stronger effect is another apparent "banding" of greatly enhanced water vapor (>2 g/kg) at plume top and a somewhat weaker water vapor decrease (~1 g/kg) at the base of the plume. Strong anomalies right at the inversion are due to changes in MBL height between the cases, not FT plume effects.

The third row of Figure 5 shows the curtains for the FireOn-RadOff differences (theoretically representing the effects of smoke radiative effects alone) and the fourth row shows the curtains for the RadOff-FireOff differences (theoretically representing the effects smoke microphysical effects alone). For the FireOn-RadOff differences, there is a caveat as smoke microphysical brightening of MBL clouds can enhance absorption in the overlying smoke (Saide et al., 2015), and thus there is an interaction between the semi-direct and indirect effects. Absolute values for the RadOff trajectory are provided in Figure

S2g-i.

        The FT temperature and moisture differences between FireOn and RadOff are substantial but also nearly identical to those between FireOn and FireOff. As a corollary, FT differences in the thermodynamic fields between FireOff and RadOff are negligible. We can thus conclude that the FT thermodynamic effects, including the "banding" features, are primarily caused by smoke radiative effects. Because comparing both FireOn-FireOff and FireOn-RadOff FT fields would therefore be

redundant, we simplify the following analysis by focusing on only one or the other combination.

        The differences in aerosol number between FireOn and RadOff (Figure 5i) appear to be mainly due to a change in location of the plume between the runs, with the plume in RadOff coming into contact with the MBL several hours sooner than in FireOn and disappearing completely by the end of the trajectory (rather than just petering out into a small remnant). As a result, the MBL is generally less smoky in FireOn than it is in RadOff until near the end of the trajectory.

There are also differences in dynamical fields around the trajectories between the runs, shown in Figure 6. The horizontal and vertical winds shown are averaged as described in Section 2.2.1. Compared to the FireOff case, FireOn shows an evolving pattern of southerly to northeasterly anomalies above the MBL. The vertical velocity fields are extremely noisy (Diamond, 2020), but the average subsidence profiles (Figure 6e) reveal an overall tendency for reduced subsidence (anomalous uplift) in FireOn as compared to FireOff. The FireOff and RadOff dynamic fields are very similar, as for the

thermodynamic fields, and thus only the FireOn-FireOff differences are shown. The exact mechanisms by which smoke radiative effects influence the FT thermodynamic and dynamic fields are explored in greater detail in Section 3.2.

        The wind shear apparent between the MBL and FT and within the FT in Figure 6 also reveal that though it is tempting to view the aerosol evolution in Figure 5 as a single plume subsiding down in time, in reality the plumes are not moving coherently with the MBL flow. Figures S3 and S4 show the plume evolution (as measured by black carbon concentration) in

the horizontal through time (averaged between 2 and 5 km) and time-mean structure with altitude, respectively. The north- and westward displacement of the plume at low altitudes (within the MBL) as compared to within the lower FT (1500 to 2500 m) is due to the aforementioned issue of MBL entrainment timescale (Diamond et al., 2018) and differences in advection between the relatively steady MBL flow and more variable FT flow.








**Figure 5.** Lagrangian curtains of liquid water potential temperature, total water mixing ratio, and accumulation mode aerosol number concentration from 20:00 UTC on 15 August 2017 (doy 227) to 20:00 UTC on 18 August 2017 (doy 230). (a-c) show the FireOn case alone, (d-f) show the FireOn-FireOff difference, (g-i) show the FireOn-RadOff difference, and (j-l) show the RadOff-FireOff difference. Black contours indicate FireOn $N_a$ of 1000 mg$^{-1}$ and grey contours indicate RadOff $N_a$ of 1000 mg$^{-1}$.



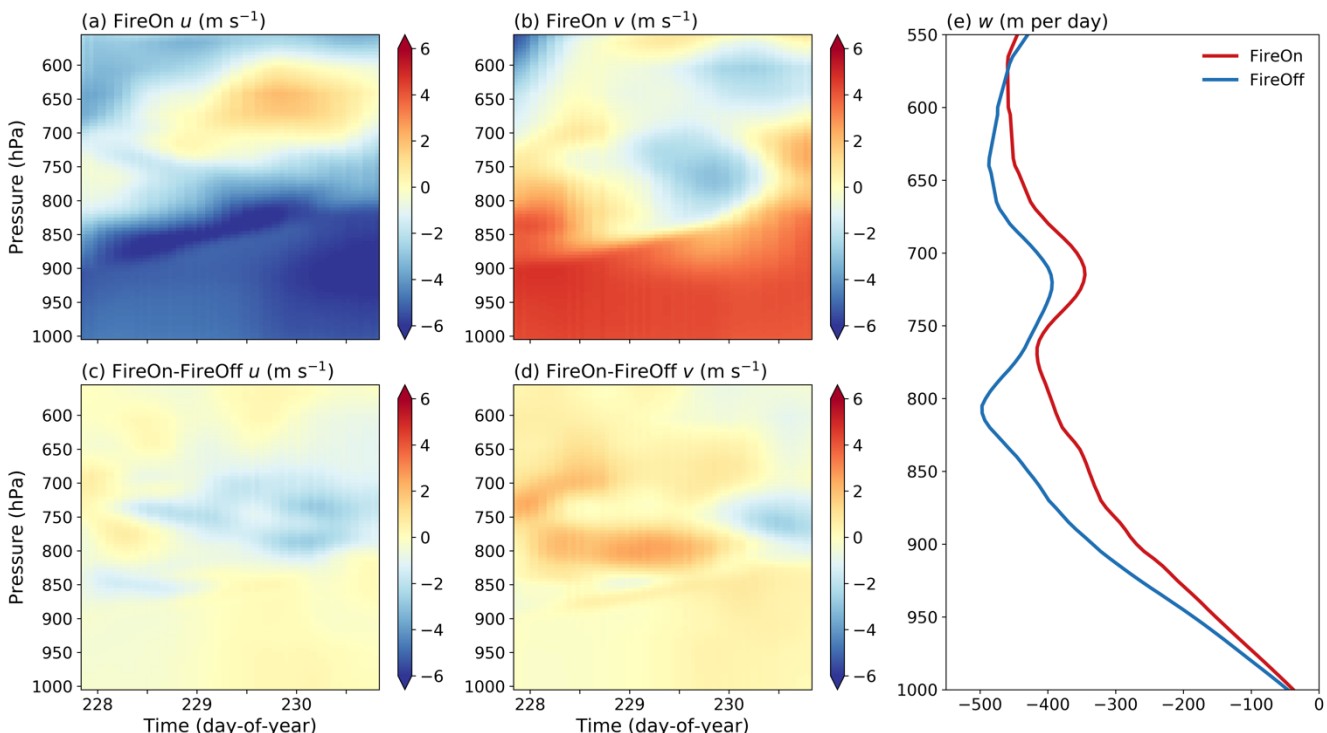

**Figure 6. Lagrangian curtains of horizontal winds and averaged profiles of subsidence. Zonal and meridional winds are shown for (a-b) FireOn and (c-d) the FireOn-FireOff difference, respectively. The (e) subsidence profiles are averaged over all times.**

Figure 7 shows the evolution of various low cloud (defined as below 3 km) and MBL properties over the course of the trajectory (again considering a 3º x 3º region centered at the trajectory location) for the three runs. For liquid water path (vertical integral of liquid water content) and $N_c$ (which is taken as the vertical average of in-cloud values weighted by liquid water content), grid boxes are considered "in-cloud" if they have a cloud fraction greater than 0.05.

Conditions are almost completely overcast throughout for FireOn, FireOff, and RadOff (Figure 7a). In general, WRF-CAM5 appears to reproduce cloud fraction in the stratocumulus regions but greatly overproduces marine clouds towards the broken cloud region as compared to SEVIRI observations (Figure S5; also shown in Doherty et al., 2022). Thus, it is not possible to assess the SCT using cloud fraction as a primary metric for the WRF-CAM5 simulations.

The MBL in all cases deepens by approximately one kilometer over the three-day evaluation period, broadly as expected for the transition over increasing sea surface temperatures (Figure 7b). RadOff and FireOff deepen a couple of hundred meters more than in FireOn by the third day, likely due to the stronger inversion in FireOn from smoke absorption (Figure 7i), at least for the majority of the trajectory (the banding effect leads to a weaker inversion on the final day).





Cloud droplet number concentration is substantially enhanced in FireOn and RadOff as compared to FireOff (Figure 7c). However, it is worth noting that the clouds are fairly polluted by remote oceanic standards (typically ~100 mg$^{-1}$ or fewer) even in FireOff. Although FireOff does not include aerosol from African biomass burning, other sources of natural (e.g., dust) and anthropogenic (e.g., urban) continental aerosol are still present. RadOff clouds have somewhat higher $N_c$ than those in FireOn, consistent with the increase in MBL aerosol number for most of the study period due to the "head-start" on smoke

entrainment in RadOff (Figure 7d).

        FireOn liquid water path exceeds that of RadOff and FireOff on the first two days but is substantially lower on the third day (Figure 7e). This timing corresponds to the switch between smoke being primarily above-cloud (first and second days) versus below-cloud (third day), and thus would be consistent with a standard understanding of semi-direct effects with above-cloud absorption favoring and below-cloud absorption disfavoring cloudiness (Johnson et al., 2004). The relative

strength of the inversion jumps between FireOn and FireOff/RadOff (greater in FireOn when the core smoke plume is above-cloud) supports this interpretation (Figure 7i). Precipitation does appear to be suppressed in RadOff and especially FireOn compared to FireOff on the third day (Figure 7f). In FireOn this is likely a combination of microphysical effects and the lower LWP whereas the RadOff precipitation suppression can be attributed to microphysics more explicitly. This may also explain the slightly higher LWP in RadOff as compared to FireOff on the third day (~200 g m$^{-2}$ versus ~175 g m$^{-2}$, respectively). Cloud

optical thickness largely follows the liquid water path (Figure 7g), with a spurious diurnal cycle due to WRF-CAM5 not defining cloud optical thickness at night.

        There is no consistent change in MBL decoupling between the runs (Figure 7k), despite FireOff and RadOff deepening more than FireOn. On the third day, it is possible that the effect of smoke absorption within the MBL in FireOn, which would generally be expected to enhance decoupling (Zhang and Zuidema, 2019), may be compensated by the lack of

evaporation from precipitation, and vice versa for FireOff (which has an influence on decoupling from precipitation but not from smoke absorption). RadOff, however, should not be affected much by either the evaporation of precipitation (very low rain water path) or smoke absorption but has essentially identical decoupling parameter values, so caution is warranted in interpreting any decoupling differences and the processes responsible. There are similarly no consistent changes in MBL-average turbulent kinetic energy between the runs.






**Figure 7. Various cloud, aerosol, and MBL characteristics over the course of the trajectory for each WRF-CAM5 case. Lines represent means and shading interquartile ranges of (a) cloud fraction, (b) inversion height, (c) cloud-weighted $N_c$, (d) MBL-average $N_a$, (e) in-cloud liquid water path, (f) rain water path, (g) cloud optical thickness, (h) MBL-average turbulent kinetic energy, (i) the inversion jump in liquid water potential temperature, (j) above-cloud relative humidity, (k) a decoupling parameter, (l) turbulent surface fluxes (latent plus sensible), (m) shortwave cloud radiative effect, and (n) longwave cloud radiative effect for FireOn (red), FireOff (blue), and RadOff (pink). White backgrounds indicate daytime and gray nighttime.**



Differences in cloud micro- and macrophysical evolution lead to differences in shortwave (SW) and, to a lesser extent,
longwave (LW) cloud radiative effects (CRE; defined as all-sky versus clear-sky flux differences) between the three cases
(Figure 7m-n). Especially on the third day, the substantially more negative shortwave cloud radiative effects in FireOff and
RadOff as compared to FireOn align well with the much larger liquid water paths maintained in the former cases. Although
the more negative cloud forcing in RadOff as compared to FireOff is likely due to both greater LWP and enhanced $N_c$ (Twomey
effect) on the third day, the more negative cloud forcing for RadOff on the first day is likely due to the Twomey effect alone,
as liquid water paths are similar between RadOff and FireOff. The less negative cloud radiative effect for the FireOn case as
compared to RadOff on the first day is somewhat surprising given that the FireOn clouds have similar concentrations of cloud
droplets and greater liquid water paths on average during that day but may be explained via aerosol direct effects. In clear-sky
conditions, smoke produces a negative radiative forcing due to the relative darkness of the underlying ocean surface, whereas
in cloudy conditions, smoke should produce a positive forcing due to the brighter underlying clouds. Thus, the difference
between top-of-atmosphere net radiation for clear-sky and overcast conditions is affected by the changing magnitude and sign
of the aerosol direct radiative effect as well as by semi-direct and indirect effects. Interpretations of cloud radiative effects in
regions with overlying absorbing aerosols thus must account for direct aerosol effects in addition to changes in cloud properties
to avoid potentially misleading conclusions (Ghan, 2013). The more positive longwave cloud radiative effects in FireOff and
RadOff as compared to FireOn are consistent with the greater liquid water paths and higher cloud tops in FireOff and RadOff.

**Section 3.2 Large-scale smoke-circulation interactions**

To determine the cause of the differences in the Lagrangian curtains — particularly the non-intuitive "banding" effects
(Figure 5) — it is helpful to look at the vertical profile of the thermodynamic and smoke profiles in more detail. Figure 8 shows
a snapshot of the vertical profiles of liquid water potential temperature, total water mixing ratio, and accumulation mode
aerosol number concentration at 21:00 UTC on 16 August 2017 (doy 228), which is approximately the time at which contact
is established between the base of the smoke plume and top of the MBL in the FireOn case. In the $\theta_l$ field (Figure 8a), all three
cases show weakly decoupled 1-1.5 km deep MBLs capped by fairly strong inversions below what looks like another relatively
well-mixed (vertically uniform in $\theta_l$) layer in the free troposphere from ~1.5-2.5 km capped by another inversion, followed by
a typical subtropical FT profile ($\theta_l$ decreasing with decreasing altitude because of radiatively-driven subsidence) above ~3 km.
The inversion layer between the neutrally and stably stratified free tropospheric air masses is located several hundred meters
higher in altitude in the FireOn case than in FireOff or RadOff. The neutrally stratified layer in the FT capped by an inversion
is also moister (Figure 8b) and contains more aerosol (Figure 8c) than the stably stratified layer above. We infer that the
neutrally stratified FT airmass represents air that was once part of the south-central African continental boundary layer (CBL)
and has since been advected offshore. Thus, this airmass will be referred to as the ex-continental boundary layer, or ex-CBL,
in the ensuing discussion. The ex-CBL is warmer, (somewhat) moister, and, perhaps most intriguingly, higher in altitude when
smoke-radiation interactions are included (FireOn) than without (FireOff and RadOff).







**Figure 8. Snapshot of smoke contact and schematic of idealized smoke displacement mechanism. Vertical profiles of (a) liquid water potential temperature, (b) total water mixing ratio, and (c) accumulation mode aerosol number concentration for the WRF-CAM5 FireOn (red), FireOff (blue), and RadOff (pink) trajectories at 21:00 UTC on 16 August 2017 (13.6° S, 4.8° W). Solid lines represent means and dashed lines the interquartile range. (d) Schematic of idealized thermodynamic profiles of "smoky" (FireOn; red) and "clean" (FireOff/RadOff; blue) ex-CBLs displaced vertically.**





A schematic (Figure 8d) of idealized ex-CBLs, one from a warmer and moister "smoky" environment and the other from a cooler and drier environment wedged between stably stratified layers and displaced vertically by a couple of hundred meters, can help explain the banding effects seen in Figure 5. For maximum simplicity, the ex-CBL is assumed to be completely well-mixed in thermodynamic properties. Observed and simulated (Figure 8a-c) ex-CBL structures are relatively well mixed, perhaps aided by longwave cooling associated with the sharp moisture gradient at plume top (Zhang and Zuidema, 2021). The "smoky" (FireOn) ex-CBL is warmer and slightly moister than the "clean" (FireOff/RadOff) ex-CBL and displaced higher vertically. (RadOff is considered "clean" despite having smoke particles because, for the purposes of the dynamic and thermodynamic changes of interest, the smoke particles themselves and their microphysical effects must play a negligible role, at least in the WRF-CAM5 simulation.) The non-CBL, stably stratified FT air is assumed not to vary between the cases.

Figure 8d is able to recreate the main features of the thermodynamic differences from the FireOn-FireOff and FireOn-RadOff curtains in Figure 5. Within the "core" of the plume, temperature is elevated and moisture is slightly higher in FireOn. At the top of the plume, the apparent small cooling and large "band" of moistening is the result of the vertical displacement, not a physical cooling or input of moisture. At the base/below the plume, the temperature effects taper off to zero while the apparent band of moderate drying is similarly an artifact of the vertical displacement. Even if the in-plume moisture enhancement were a fluke, the "banding" effects of apparent moistening at plume top and drying at plume bottom would still apply. While these effects may not be "physical" in the sense that they do not require the input or loss of water in a plume-relative frame, they are "real" in the sense that the clouds respond to the differences in geometric space and thus the vertical shift matters for physical MBL processes.

What causes the vertical displacement of the ex-CBL between FireOn and FireOff/RadOff? The answer must somehow involve aerosol-radiation interactions, as those are the only smoke effects that should be near-identical between the FireOff and RadOff simulations. One possibility is that the CBL is deepened over land in the presence of aerosol-radiation interactions due to enhanced convection via an "elevated heat pump" mechanism in which bottom-heavy smoke absorption warms and destabilizes the mid-troposphere (Allen et al., 2019). In that scenario, the CBL differences would originate over land and simply be advected out over the ocean. Alternatively, the CBL differences could originate in transit over the ocean as smoke absorption reduces large-scale subsidence (Adebiyi et al., 2015; Das et al., 2020; Das et al., 2017; Mallet et al., 2020; Sakaeda et al., 2011).

Figure 9 shows the WRF-diagnosed planetary boundary layer height (PBLH) above the local surface for FireOn and the FireOn-RadOff difference at 12 UTC averaged over all days (14-20 August 2017). Contrary to the hypothesis of enhanced convection over land, boundary layer deepening is actually suppressed over land due to smoke-radiation interactions. Longitude-altitude curtains at 10º S of the FireOn-RadOff liquid water potential temperature and total water mixing ratio differences averaged over all times show that smoke radiative effects shade and cool the surface while warming the mid-troposphere (Figure 9c). The stabilization of the lower troposphere suppresses convection in this case, similar to previous results from northern Africa (Tosca et al., 2015). The near-surface moistening and drying aloft (Figure 9d) can be explained as a consequence of the CBL shallowing driven by the stabilizing temperature changes. Less free tropospheric entrainment



leads to more moisture remaining at low altitude without being diluted and, by the same token, less near-surface moisture transport to higher altitudes. The effects appear to maximize at sundown (after a full day of solar absorption) and decay, although not disappear entirely, overnight (Diamond, 2020).


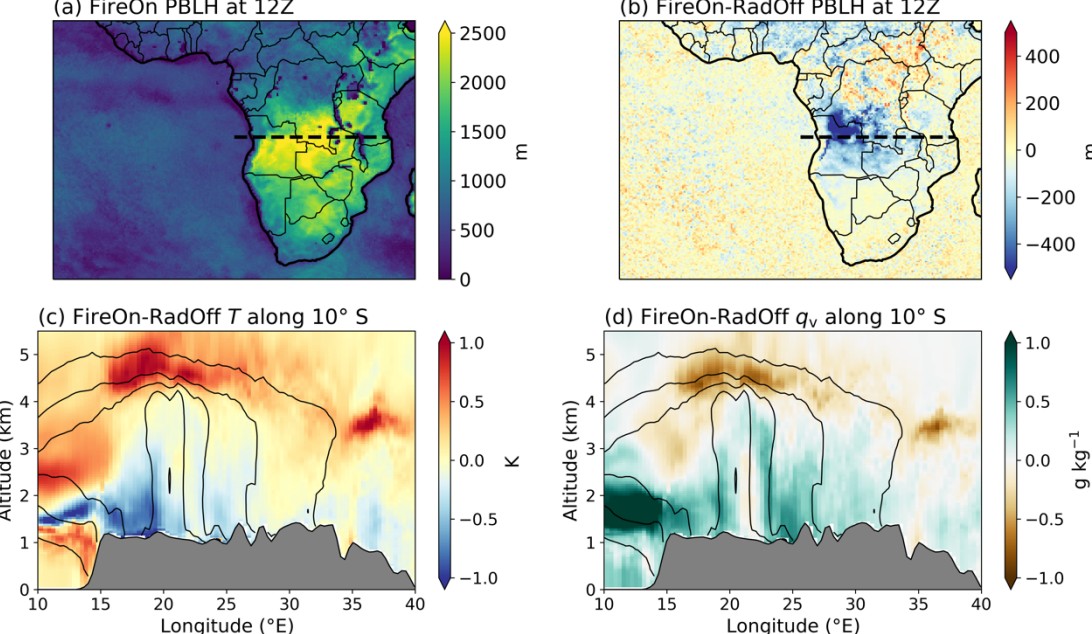

**Figure 9. Effects of smoke absorption on continental convection.** WRF-diagnosed planetary boundary layer height at 12 UTC for (a) FireOn and (b) the FireOn-RadOff difference over the full domain (40° S-15° N, 30° W-50° E). The dotted lines in (a-b) indicate the location of the profiles (along 10° S from 10-40° E) of FireOn-RadOff (c) temperature and (d) specific humidity. Thermodynamic
differences in (c-d) are averaged over 14-20 August 2017 at all times. Black contours represent black carbon mass concentration at intervals of 0.5 μg kg⁻¹. Grey shading represents the continental landmass.

Changes in subsidence driven by smoke absorption over the ocean is thus left as the most likely explanation for the vertical displacement in the ex-CBLs. We can evaluate the effect of diabatic heating on subsidence by expressing the First
Law of Thermodynamics as:

$$Q - \vec{u} \cdot \nabla(T) = \frac{\partial T}{\partial t} + (\Gamma_d - \Gamma_e)w, \tag{2}$$

where $Q$ is the diabatic heating rate, $\vec{u}$ is the horizontal wind vector, $T$ is temperature, $\Gamma_d$ is the dry adiabatic lapse rate, $\Gamma_e$ is the environmental lapse rate, and $w$ is the vertical velocity. Energy inputs via diabatic heating or temperature advection must be balanced by changing temperature (temperature tendency term) or performing work (stability x vertical velocity term).
Although $Q$ was not explicitly included in the WRF-CAM5 output, it can be calculated as the residual from the other terms, all of which can be calculated from the standard output variables.





Figure 10 shows the difference in the column atmospheric radiative heating rate ($Q_R$) and profiles of subsidence over the SEA from FireOn and RadOff and Figure 11 shows the breakdown of the terms in Eq. (2) averaged between 2 and 5 km and over the full 14-20 August 2017 period. Anomalous radiative heating from smoke over the ocean in FireOn is associated

with subsidence rates ~100 m per day (~25-40%) weaker than in RadOff. Because the Coriolis force is weak in the tropics and strong horizontal temperature gradients cannot be maintained, we should expect the effects of a radiative heating to be expressed more as a thermally-driven circulation change than as a temperature change (Dagan et al., 2019). This expectation is borne out in the daily average FireOn-RadOff differences in Figure 11.

We can also divide the effects into separate daytime and nighttime averages to confirm that our interpretation is

correct — or rather not clearly incorrect, as would be the case if the bulk of the apparent diabatic heating occurred at night. Consistent with our expectations, the entirety of the diabatic heating enhancement occurs during the day. If there were large moisture enhancements over the entire plume area due to smoke-radiation interactions, we may also have expected to see an anomalous diabatic cooling at night. During the day, much of the anomalous diabatic heating goes into warming the air, with a substantial portion acting to reduce subsidence as well. At night, the reduction in subsidence is balanced by an apparent

cooling tendency. This can be explained more straightforwardly as a reduction in compression warming in the more weakly subsiding air than as a direct cooling effect. Reductions in both static stability and subsidence play a role in setting the strength of the stability/subsidence term, although the decrease in subsidence dominates over most of the smoky region (Diamond, 2020).

Reduced subsidence over the ocean, rather than (or in opposition to) differences in CBL height over land, is thus

responsible for the elevated ex-CBL altitude in FireOn as compared to FireOff and RadOff and the resulting thermodynamic "banding" effects apparent in the Lagrangian curtains.

Low-level horizontal winds are also affected by the large-scale smoke-circulation interaction. Figure 12 shows average anomalies (calculated as the FireOn-RadOff difference) in sea level pressure (SLP) and horizontal winds at 2 km for the full 14-20 August 2017 period. The smoke absorption and anomalous lifting set up a heat low and cyclonic circulation

pattern on average, although instantaneous snapshots of FireOn-RadOff differences are more variable. The horizontal wind differences just above the MBL in Figure 12 are broadly consistent with the trajectory starting around the western flank of a cyclonic anomaly (anomalous southerly wind). The cyclonic anomaly produced during just one week is strikingly similar to that produced seasonally in a multi-year regional climate model study of the SEA (Mallet et al., 2020).






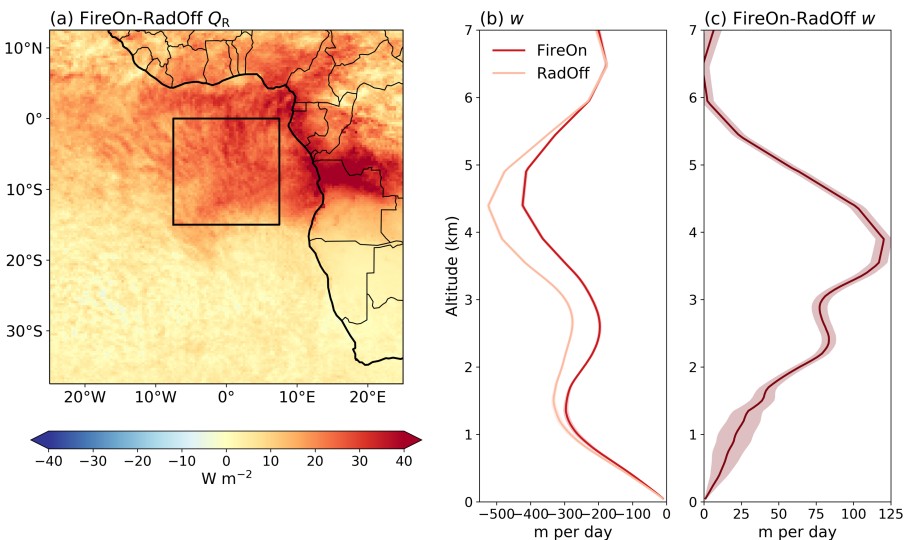

**Figure 10. Atmospheric column radiative heating and vertical profile of subsidence differences between FireOn and RadOff. Subsidence values averaged over the region from 0° to 15° S, 7.5° W to 7.5° E and over the full 14-20 August 2017 period, demarcated in black in (a), are shown separately for FireOn and RadOff in (b) and as the FireOn-RadOff difference in (c). Shading in (b-c) represents two standard errors of the mean.**

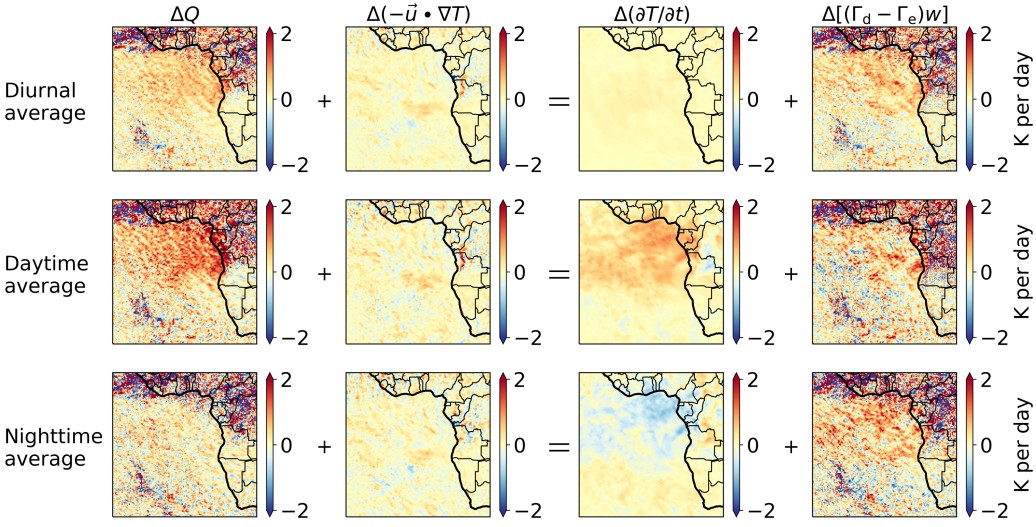

**Figure 11. Decomposition of terms in the First Law of Thermodynamics averaged between 2 and 5 km and over the full 14-20 August 2017 period. FireOn-RadOff differences in diabatic heating (first column), temperature advection (second column), temperature tendency (third column), and the product between stability and vertical velocity (last column) for the diurnal average (top row), daytime only (middle row), and nighttime only (bottom row).**



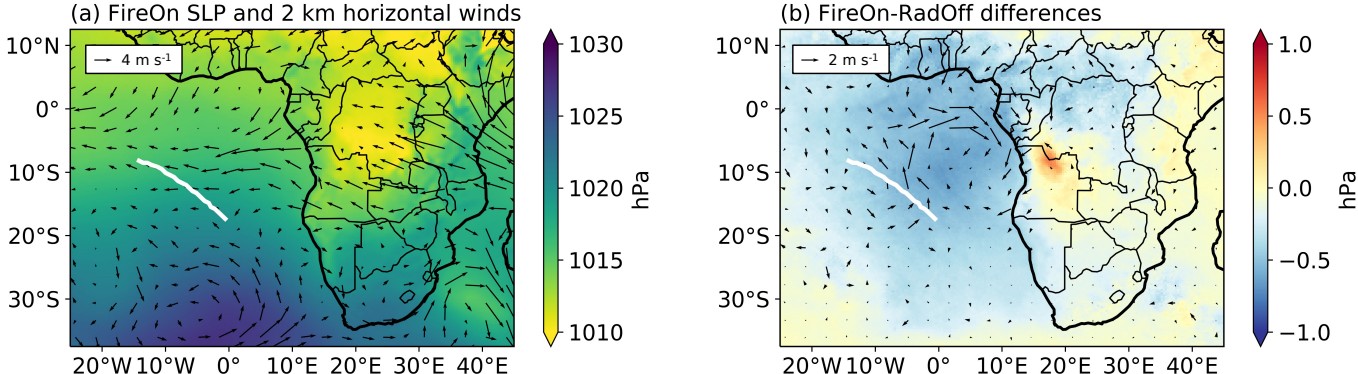

**Figure 12. FireOn (a) and FireOn-RadOff differences in (b) sea level pressure and horizontal winds at 2 km over the full 14-20 August 2017 period. SLP is shown as shading and wind as quivers. Note the different arrow length scaling in (a) and (b) for horizontal winds and their differences, respectively. The white line represents the Lagrangian trajectory location.**

The dynamical changes due to smoke absorption have implications for cloud microphysics as well. Figure 13 shows maps of the average (in-cloud) cloud droplet number concentration for marine low clouds in FireOn and the FireOn-FireOff, FireOn-RadOff, and RadOff-FireOff differences. The $N_c$ enhancement is greatest north of 10° S. The peak values within the smoke-affected area are much higher in WRF-CAM5 than seen in the satellite climatology (Grosvenor et al., 2018), although this may be due to the fact that we are only analyzing conditions in one week whereas the climatology encompasses multiple years of data. The spatial distribution of $N_c$ in the Eulerian sense tracks the evolution of the FireOn and RadOff trajectories in general. The lower droplet concentration in FireOn as compared to RadOff is consistent with the Lagrangian perspective and with the explanation that smoke plumes closer to cloud top have more time and opportunity to be entrained into the MBL. Thus, the reduction in subsidence driven by smoke absorption is not only relevant in terms of the semi-direct effect, but also influences the timing and magnitude of indirect effects as well by modulating smoke-cloud contact.

Aerosol-radiation interactions, therefore, can account for both the thermodynamic and dynamic FT differences between the runs. Importantly, previous LES studies like those of Y15 and Z17 could not fully account for such effects because the domain is too small to capture these large-scale circulation interactions and their forcings were not designed for the southeast Atlantic. Additionally, climatological or composite trajectories from the southeast Atlantic may also produce misleading results as the "banding" effects that could be important for an individual case would be diluted by averaging across many situations.





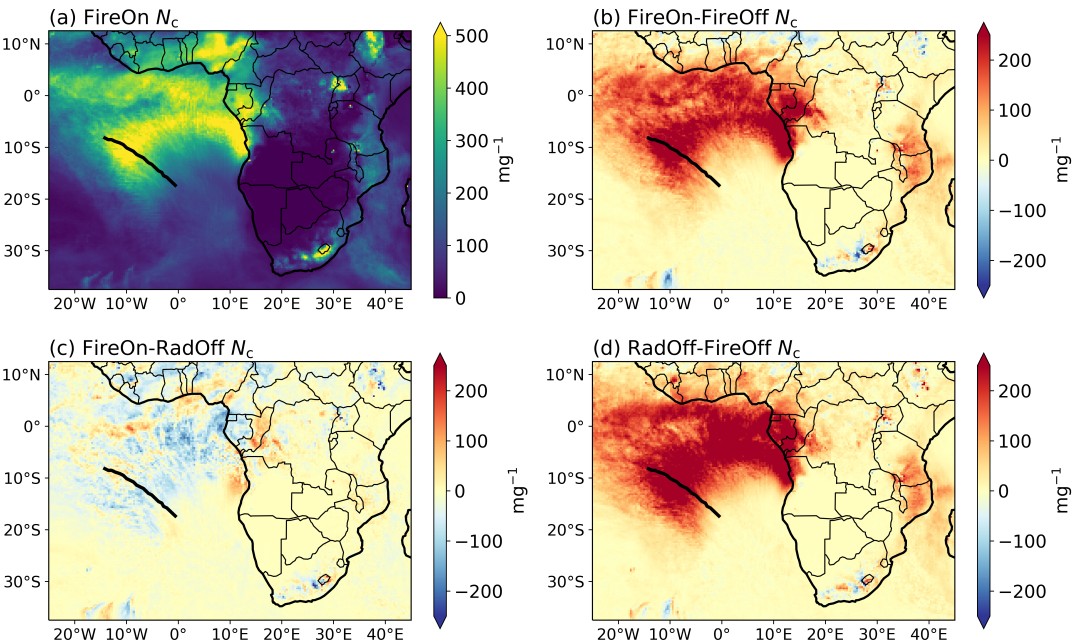

**Figure 13. Cloud droplet number concentration differences for the full simulation period. Average cloud-weighted $N_c$ is shown for (a) FireOn and the (b) FireOn-FireOff, (c) FireOn-RadOff, and (d) RadOff-FireOff differences. The black line represents the Lagrangian trajectory location.**

## 4 Large eddy simulation results

A summary of aerosol and cloud microphysical and macrophysical evolution for the base SAM cases using forcing fields taken entirely from WRF-CAM5 FireOn (AllOn) and FireOff (AllOff) is shown in Figure 14. The transition from single-layer stratocumulus clouds on the first day to a cumulus-under-stratocumulus or shallow cumulus configuration by the final day is apparent in both simulations. With smoke included, the MBL grows progressively more polluted over time and cloud droplet number concentration increases, especially after the "core" of the plume comes into contact with the cloud tops during the second night. (A cloud fraction threshold of 1% and cloud water threshold of 0.1 kg kg$^{-1}$ is used to define "in-cloud" for the purposes of visualizing $N_c$ in Figure 14 to better include values for the scattered cumulus clouds.) Peak MBL pollution and cloud droplet number are not coincident with peak above-cloud smoke concentrations because the entrainment process has a characteristic timescale on the order of days (Diamond et al., 2018). A similar phenomenon (albeit with much lower aerosol number) is evident for MBL pollution in the AllOff case as well. The cloud field remains mostly overcast with stratiform clouds until the end of the AllOn simulation, whereas breakup into scattered cumuliform clouds is evident midway through the AllOff simulation.


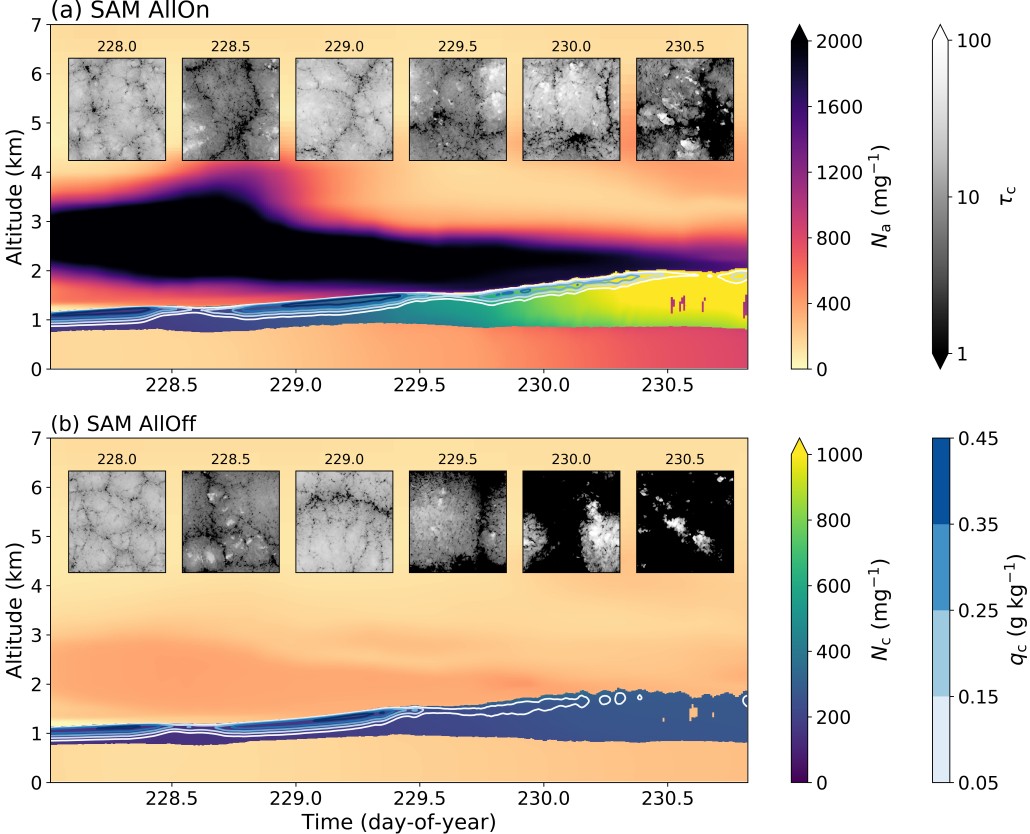

**Figure 14. Overview of microphysical and macrophysical cloud evolution for the base AllOn and AllOff cases. The evolution of domain-average aerosol (orange to purple shading) and cloud droplet (blue to yellow shading) number concentration and cloud water mixing ratio (blue contours) are shown over the course of the Lagrangian simulation for the (a) AllOn and (b) AllOff cases. Insets show the plan-view of cloud optical thickness fields every 12 hours. Colorbars are shared between (a) and (b).**

The evolution of various cloud and MBL properties over the course of the Lagrangian simulations for AllOn and AllOff along with the microphysics-focused and the local semi-direct effect-focused sensitivity tests (NOff, AllOff$_{N/2}$, AsiAbs) and the large-scale semi-direct effect focused sensitivity tests (TOff, QOff, WOff) are shown in Figure 15 and Figure 16, respectively.

Cloud in AllOn breaks up more than in AllOff during the first day, but fully recovers to overcast conditions during the second night and maintains consistently higher cloud fraction than AllOff from the second day onward (Figure 15a). AllOn deepens at a slightly faster rate than AllOff (Figure 15b). Both simulations start off with $N_c$ values of approximately 200 mg$^{-1}$ (Figure 15c). AllOn entrains aerosol efficiently and produces extremely polluted clouds with $N_c$ values exceeding 1000 mg$^{-1}$ by the end of the simulation (which is in line with the final $N_c$ values simulated in the smoky cases in Y15 and Z17). AllOff maintains relatively steady levels of cloud droplet and aerosol number concentrations, indicating that a precipitation-feedback driven transition as simulated in Y15 and Yamaguchi et al. (2017) is not driving the differences between the case AllOn and





AllOff cases. Indeed, neither simulation produces appreciable rain water (Figure 15f). Instead, the weaker inversion (Figure 15i) and (at least toward the end of the simulation) drier free troposphere (Figure 15j) are more plausibly tied to the cloud fraction evolution, although the strength of the MBL decoupling does not differ greatly (Figure 15k). Surface fluxes (Figure 15l) are greater in AllOff, which is consistent with the more trade cumulus-like ending state. Shortwave and longwave cloud radiative effects (Figure 15m-n) largely track the cloud fraction differences.

The differences in SCT evolution between AllOn and NOff in Figure 15 can be interpreted as the result of indirect effects alone (notwithstanding the caveat that larger-scale circulation adjustments may affect smoke location and entrainment). Perhaps the most striking result is the lack of strong differences between AllOn and NOff in terms of cloud fraction or deepening (Figure 15a-b), despite the five-fold enhancement in $N_c$ and $N_a$ in AllOn compared to NOff (Figure 15c-d).

         One explanation for the lack of a drizzle-driven transition is that even in AllOff, the background accumulation mode
aerosol concentrations in the free troposphere would not be considered "clean" by typical remote marine standards. Indeed, even though there is no smoke, it is possible to see an enhancement in aerosol within the ex-CBL air as compared with the rest of the free troposphere (which is sourced from subsiding air from the Hadley circulation or Southern Ocean) in Figure S2 and Figure 14b. AllOff$_{N/2}$ offers a view of a cleaner marine environment with the same thermodynamic and dynamic setup as AllOff. In this case a transition from overcast closed-cell stratocumulus to shallow open-cells occurs rapidly (Figure 15a), with
$N_c$ and $N_a$ values bottoming out around 10 mg$^{-1}$ (Figure 15c-d). (It should be noted that our simulation domain is too small to fully resolve the open-cell structure, however.) The lower cloud fraction and open-cellular mesoscale organization lead to substantially diminished entrainment (Abel et al., 2020; Kazil et al., 2021) and even boundary layer shoaling (Figure 15b). The large quantities of rain (Figure 15f) produced lead to substantially more decoupled conditions than the base AllOff case (Figure 15k). The substantially reduced cloud coverage in AllOff$_{N/2}$ leads to a greatly reduced cloud radiative effect (Figure 15m-n).
Although a detailed treatment is outside the scope of this paper, further work is merited to better understand under what conditions this type of "drizzling-depletion" transition is favored and whether such a transition can be reproduced using different LES models and microphysical schemes.

         The local (below-cloud) semi-direct effects from smoke absorption are qualitatively consistent with expectations, with the base AllOn case having greater cloud fraction than the more absorbing AsiAbs case (Figure 15a). Shortwave and
longwave cloud radiative effect differences track the cloud fraction differences (Figure 15m-n).





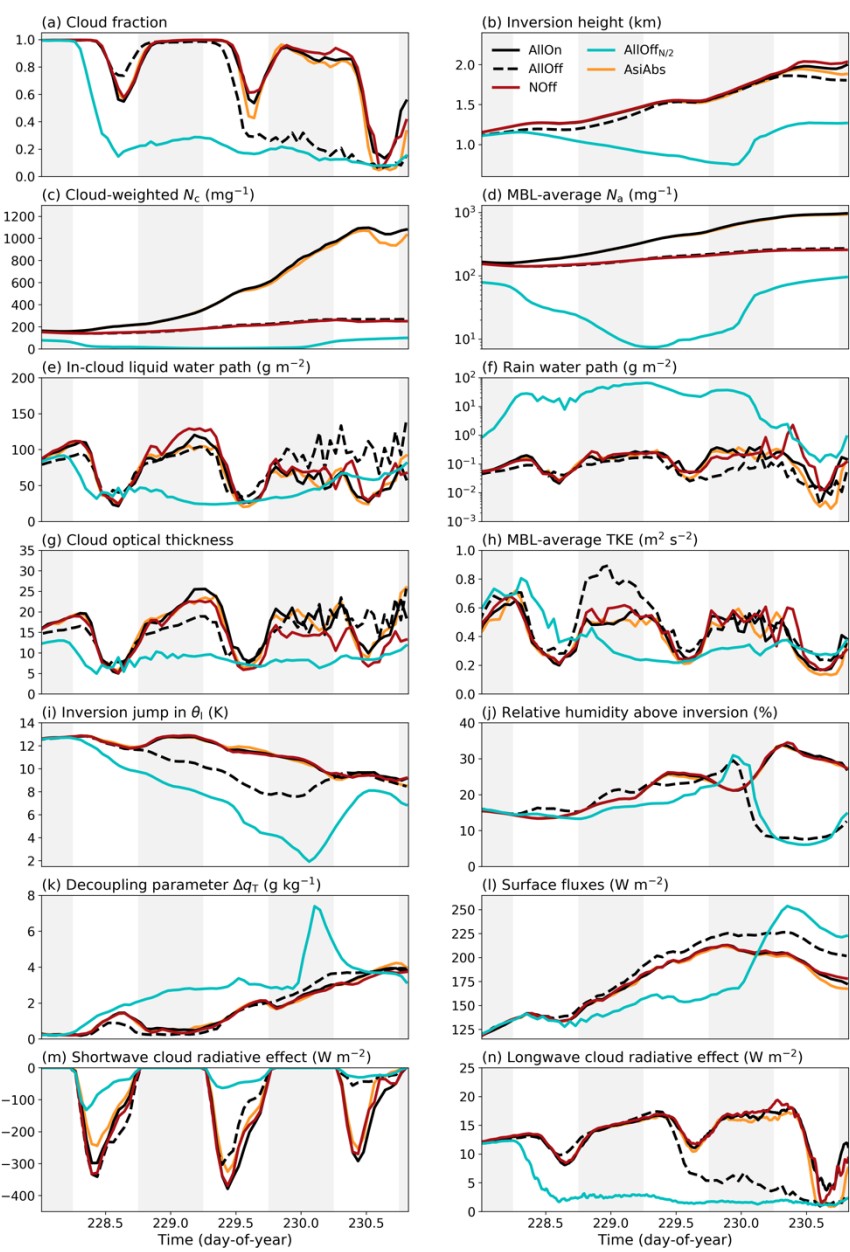

**Figure 15. Evolution of various cloud and MBL properties over the course of the Lagrangian simulations for AllOn and AllOff along with the microphysics-focused and local semi-direct sensitivity tests. Subplots are as in Figure 7 but for the LES base cases and NOff (red), AllOff_{N/2} (cyan), and AsiAbs (orange).**





**Figure 16. Evolution of various cloud and MBL properties over the course of the Lagrangian simulations for AllOn and AllOff along with the large-scale semi-direct effect focused sensitivity tests. Subplots as in Fig. 15 but for TOff (light blue), QOff (dark blue), and WOff (purple).**



The influence on the SCT from the thermodynamic and subsidence changes discussed in detail in Section 3 can be considered large-scale semi-direct effects, as opposed to the more local effects caused by cloud-layer and below-cloud absorption or the effect of absorption directly above clouds without the ability to change the larger-scale circulation as simulated in prior LES studies (Herbert et al., 2020; Yamaguchi et al., 2015; Zhou et al., 2017). The comparison of AllOn and

TOff isolates the effect of temperature alone; AllOn and QOff isolates the effect of moisture alone; and AllOn and WOff isolates the effect of subsidence alone (Figure 16).

The AllOn-TOff differences in cloud fraction (Figure 16a) follow those in inversion strength (Figure 16i), with negligible differences during the first day, shifting to a stronger inversion and higher cloud fraction in AllOn on the second day (when the warmed "core" of the smoke plume is directly above the MBL), and then finally a weaker inversion and lower

cloud fraction in AllOn on the final day (when the effect of the "banding" is now above the MBL). Consistent with the weaker inversion for most of the simulation, TOff grows more rapidly than AllOn (Figure 16b), which also leads to enhanced entrainment of smoke and thus greater $N_c$ (Figure 16c) given similar MBL turbulence (Figure 16h) and updraft speeds. The AllOn-QOff differences in cloud fraction (Figure 16a) follow those in above-cloud relative humidity (Figure 16j), with greater moisture and cloud fraction on the first day in QOff giving way to a more dramatic decline in above-cloud moisture and

concomitant cloud breakup toward the end of the simulation. Thus, the FT thermodynamic effects from smoke differ from the standard view of a stronger inversion and enhanced water vapor in the plume leading to more cloudiness but are consistent with the more subtle picture that emerges when accounting for both warming and dynamical adjustments (Figure 8).

The effects of subsidence changes are of a similar magnitude to those from the thermodynamics. WOff begins breaking up before AllOn and maintains consistently lower cloud fractions from the second day forward (Figure 16a). The

greater subsidence in WOff (Figure 6e) suppresses MBL growth by several hundred meters compared to AllOn (Figure 16b), resulting in less smoke entrainment and lower $N_c$ as well (Figure 16c-d). Interestingly, the shallower MBL in WOff is able to maintain a stronger degree of coupling than AllOn (Figure 16k), although this does not translate into greater cloud fraction.

No single variable alone is able to explain the large difference in cloud fraction between AllOn and AllOff beginning during the second day. A weaker inversion (Figure 16i) paired with stronger subsidence at the end of the second day and

beginning of the third night together appear to drive cloud fraction down and prevent nocturnal recovery while the substantially lower above-cloud humidity (Figure 16j) on the third day further precludes restoration of a thin stratocumulus layer.

Figure 17 summarizes the overall cloud fraction and broadband scene albedo changes over the course of the SCT for each simulation. All-sky albedo values are assessed within 10 bins of cloud fraction of equal width and only daytime values are included. Comparing AllOn and AllOff, scene albedo is higher in AllOn at any given cloud fraction and overall scene

albedo and cloud fraction are both higher in AllOn on average, suggesting the differences are due to changes in both cloud albedo and occurrence in the presence of smoke. The most dramatic difference with the AllOn values occurs for the AllOff$_{N/2}$ case, which is the only case in which the transition is initiated rapidly by the onset of a positive precipitation-scavenging feedback loop.



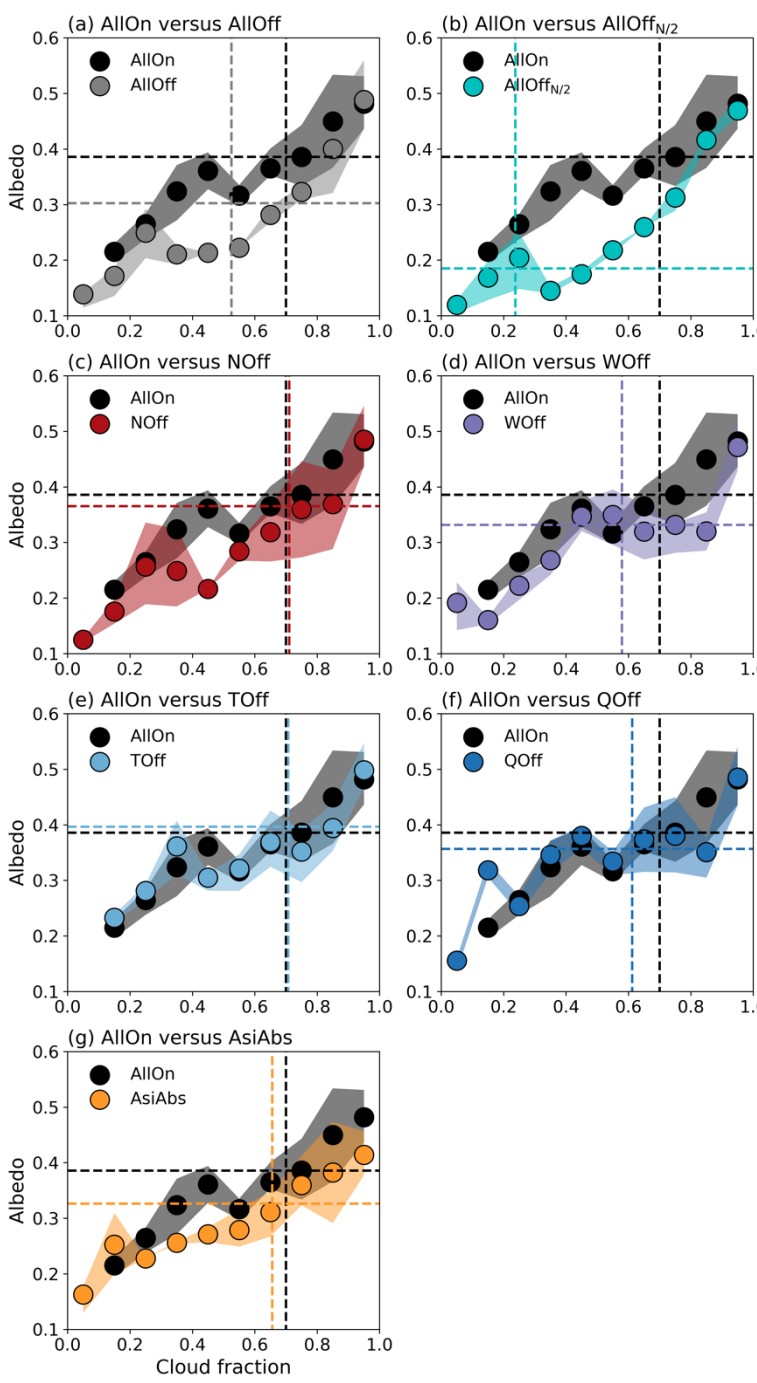

**Figure 17. Scene albedo-cloud fraction summary plots for each simulation. Binned AllOn values are compared with (a) AllOff, (b) AllOff_{N/2}, (c) NOff, (d) WOff, (e) TOff, (f) QOff, and (g) AsiAbs. Dashed lines correspond to mean daytime values over the course of the simulation and markers and shading correspond to the mean and interquartile range of scene albedo for each cloud fraction bin, respectively.**





NOff is the only simulation in which the average cloud fraction and scene albedo differences differ in sign: The Twomey effect (mainly, as liquid water changes are small or of the wrong sense in Figure 15e) compensates for the slightly lower cloud fraction in AllOn to lead to more overall cooling over the course of the transition. Indirect effects lead to the scene albedo in AllOn exceeding that of NOff for any given cloud fraction.

The effect of subsidence exceeds that of the indirect effect in this study. For the most part, scene albedo is lower in WOff than in AllOn at any given cloud fraction (both $N_c$ and LWP are persistently lower in WOff) and cloud fraction is lower in WOff overall. For TOff and QOff, scene albedo at any given cloud fraction is similar to that of AllOn and the differences in overall scene albedo are driven mainly by cloud fraction differences. The lower cloud fraction of TOff on the second day and higher cloud fraction on the third day largely cancel, leaving a small increase in cloud fraction and scene albedo averaged over the course of the SCT. This result is the opposite of what would be expected from above-cloud heating alone because of the temperature banding effect. For QOff, the lower cloud fraction on the third day more than compensates for the increased cloud fraction on the first day. The lower scene albedo in AsiAbs as compared to AllOn is from a combination of lower cloud fraction and the direct effect of enhanced above-cloud smoke absorption.

**5 Comparison with observations**

To compare the WRF-CAM5 and SAM output with observations from the 18 August 2017 ORACLES-CLARIFY joint flight, the flight period is divided into three-hour periods centered around the WRF output (12 UTC, 15 UTC, 18 UTC). We compare the WRF FireOn and SAM AllOn simulations with the observations as these simulations contain the fullest representations of the smoke radiative and microphysical effects believed to be operating in reality. WRF data is averaged over the 3º x 3º box centered along the trajectory. Aircraft data from the NASA P3 and FAAM BAe are averaged over 50 m vertical bins for the times/locations that correspond to the relevant WRF output in each period. Figure 18 illustrates this schema.

For the first time period (10:30-13:30 UTC), the two aircraft transit over the same path, descend, and begin joint sampling. The second period (13:30-16:30 UTC) begins partway through back-and-forth joint sampling along 9º S and includes the subsequent return of the FAAM BAe to Ascension Island and continued sampling by the NASA P3 along 8.5º S. The final time period (16:30-19:30 UTC) captures the descent of the NASA P3 on its return to Ascension Island.

Radiosondes launched at 12:00 and 18:00 UTC at Ascension Island as part of the LASIC campaign provide another set of thermodynamic profiles (technically potential temperature and specific humidity rather than liquid water potential temperature and total water mixing ratio) and data from the ground site is available for each time period as well.

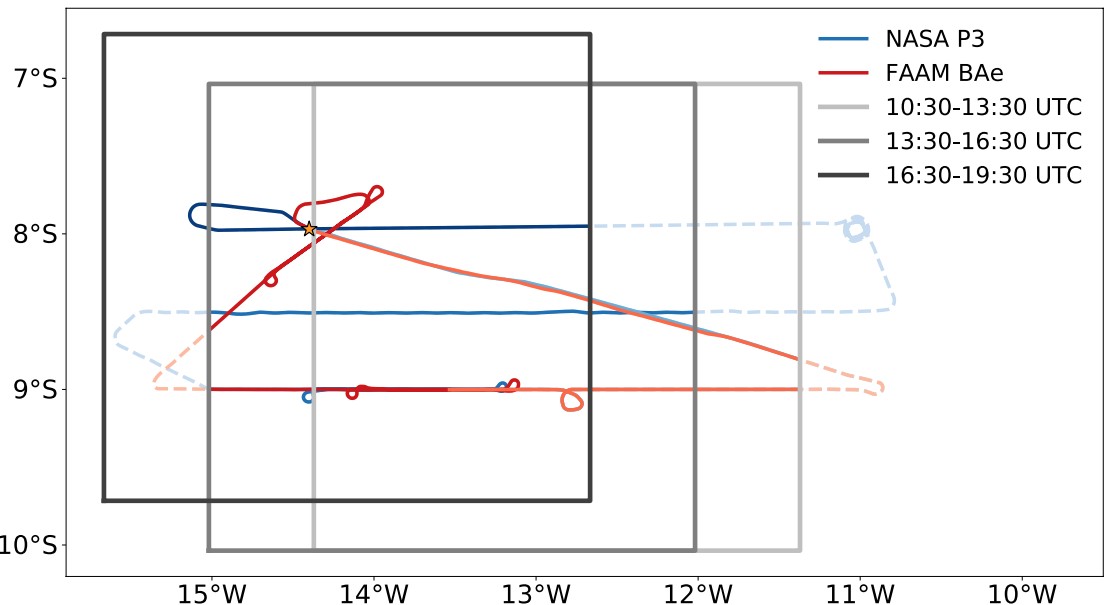

**Figure 18. Tracks for the NASA P3 and FAAM BAe aircraft during the 18 August 2017 joint flight. The regions used for the WRF model averages at each time period are indicated by gray boxes. Flight data occurring at the correct time and location for the first, second, and third time periods are indicated by light, medium, and dark colored solid lines. Dashed lines indicate flight data that does not align with any relevant model data. The location of Ascension Island is marked by the yellow star.**

Figure 19 shows temperature, moisture, and aerosol profiles from the model output and ORACLES-CLARIFY-LASIC in-situ observations for each of the three time periods analyzed on 18 August 2017. The SAM AllOn simulated MBL is deeper than that observed and, perhaps relatedly, sits under a substantially weaker inversion due to a cool and moist bias in the WRF FireOn simulated FT. The cool/moist FT bias is also a smoke bias as seen in the $N_a$ and black carbon (BC) mass concentration fields, as the models maintain a relatively narrow plume just above the MBL whereas the observations show clean air above the MBL. These observations are all consistent with what would be expected from the conceptual model in Figure 8 if the models produced a smoke plume (or ex-CBL) that extended too far toward Ascension Island/lasted too long in time. These shifts in space and time of the FT plume are to be expected in WRF-CAM5 as by this point in time it had been free-running for four days and had no constraints on smoke loading or location since the model was initialized on August 14[th].

The models also simulate too much smoke in the MBL (in terms of both accumulation mode aerosol number and BC mass) as compared to the aircraft and ground site measurements (which agree well with each other, suggesting that heterogeneity within the 3º x 3º region is an unlikely explanation). This could be a result of too much entrainment (perhaps related to the low bias in inversion strength, the prolonged plume exposure, or to numerical diffusion), an overall high bias in model smoke concentrations, or some combination.

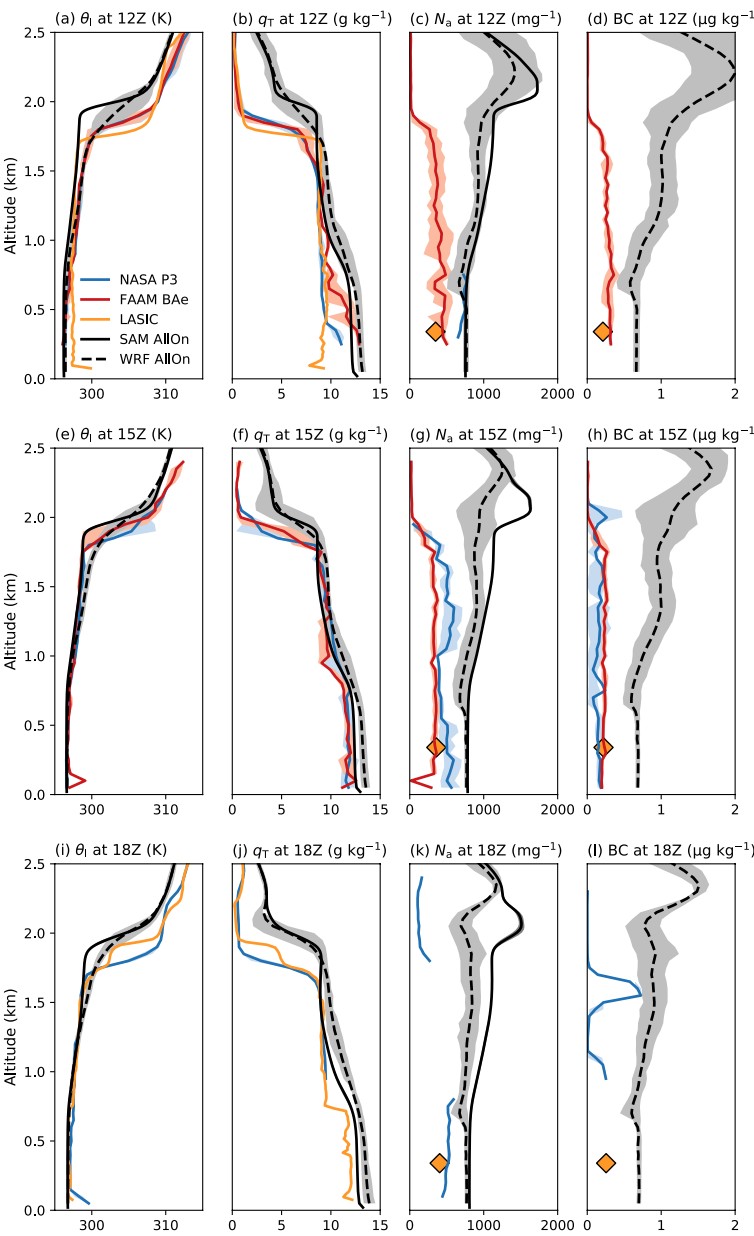

**Figure 19. Comparison of model output with co-located ORACLES, CLARIFY, and LASIC observations for three time periods. Liquid water potential temperature, total water mixing ratio, accumulation mode aerosol number concentration, and black carbon (BC) mass concentration are shown at (a-d) ~12:00 UTC, (e-h) 15:00 UTC, and (i-l) 18:00 UTC, respectively, where data is available. Lines represent the mean and shading the interquartile range at each altitude for the models, flights, and radiosondes. Ground station data is represented as a diamond marker for the mean value.**



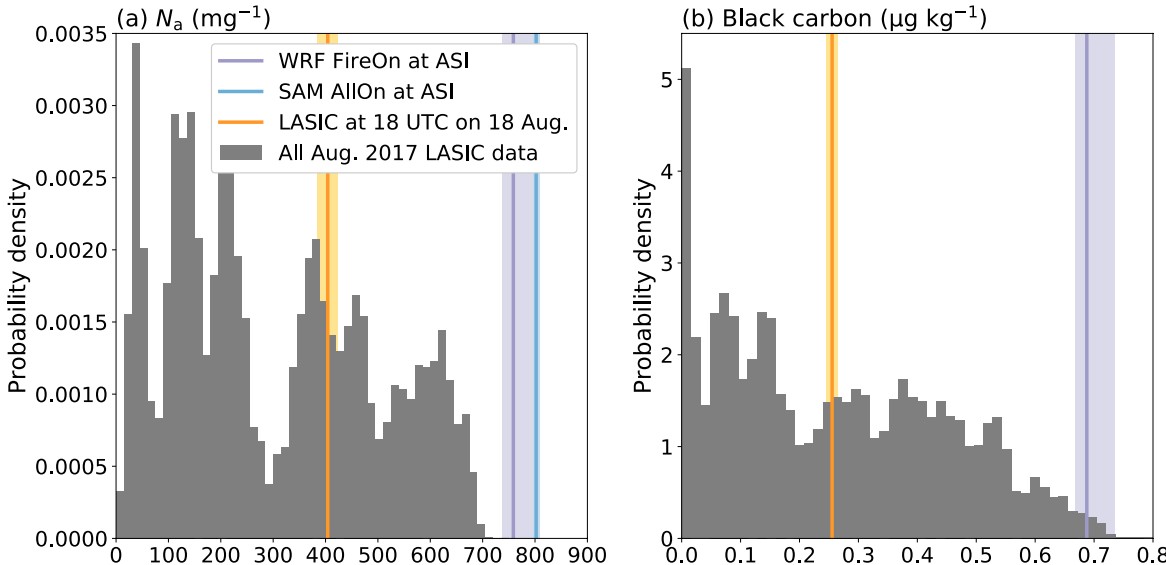

**Figure 20. Comparison of August 2017 LASIC statistics with model output. The distributions of all August 2017 observations of (a) accumulation mode aerosol number concentration and (b) black carbon mass concentration from the LASIC ground site are shown as grey histograms. Means and interquartile ranges are indicated by lines and shading, respectively, for the LASIC observations (yellow), WRF FireOn output (purple), and SAM AllOn output (blue) subset around 18 UTC on 18 August 2017.**

A broader context for the observations on 18 August 2017 is provided in Figure 20, which shows the distribution of all LASIC observations for August 2017. The smoke concentrations seen on 18 August 2017 were typical of polluted conditions measured at Ascension Island. In contrast, the simulated concentrations are either above ($N_a$) or near the high tail of (BC) the observed values for August 2017. The greater $N_a$ values in SAM AllOn as compared to WRF FireOn could be related to greater entrainment or lower coalescence scavenging in SAM. The indirect and local semi-direct effects in the various SAM runs (Figure 15) may therefore represent high-end estimates for the magnitude of aerosol-cloud-radiation interactions that characterize the SEA around Ascension Island.

Figure 21 compares the model output over the course of the full Lagrangian trajectory with geostationary retrievals from SEVIRI. WRF FireOn data is considered over the usual 3º x 3º region following the trajectory. SEVIRI data is processed in the same manner. SEVIRI cloud fraction is calculated from the cloud phase flags as the ratio of successful liquid cloud retrieval pixels to the sum of those pixels and clear-sky pixels. Pixels with different cloud phase retrievals or unsuccessful retrievals are neglected and random overlap is assumed.

SAM AllOn qualitatively matches the SEVIRI cloud fraction evolution (Figure 21a), although the breakup during the first day is exaggerated and the recovery during the final night somewhat more complete compared to the observed values. Although WRF FireOn reproduces the overcast stratocumulus and the right sense of the diurnal cycle (with breakup during



the day and nocturnal recovery) and sense of the transition (with lowest cloud fractions during the final day), it is biased toward extremely overcast conditions in the transition region (Fig. S5; Doherty et al., 2022).

$N_c$ simulated by SAM AllOn greatly exceeds that from SEVIRI and the aircraft observations (Figure 21b). This is in part due to the greater MBL $N_a$ concentration (Figure 19c,g,k), but also may represent a bias in terms of activation, as WRF FireOn also has a very polluted MBL but does not produce $N_c$ values as extreme as SAM AllOn. The high activation fractions in SAM are likely unrealistic, as observations suggest that clouds in the SEA should shift from an aerosol-limited to an updraft-limited regime at high levels of pollution (Kacarab et al., 2020). The lower SEVIRI values on the final day should be interpreted

cautiously, as the $N_c$ calculation for passive sensors is increasingly uncertain for more broken cloud scenes (Grosvenor et al., 2018).

    WRF FireOn produces consistently higher liquid water paths than SAM AllOn (Figure 21c). The SEVIRI retrievals are closer to the SAM AllOn values on the first and third day and in between the models on the second. Both models and the observations show an evolution from more normally distributed liquid water path distributions to distributions more heavily

weighted toward a tail of relatively large values as the transition progresses.

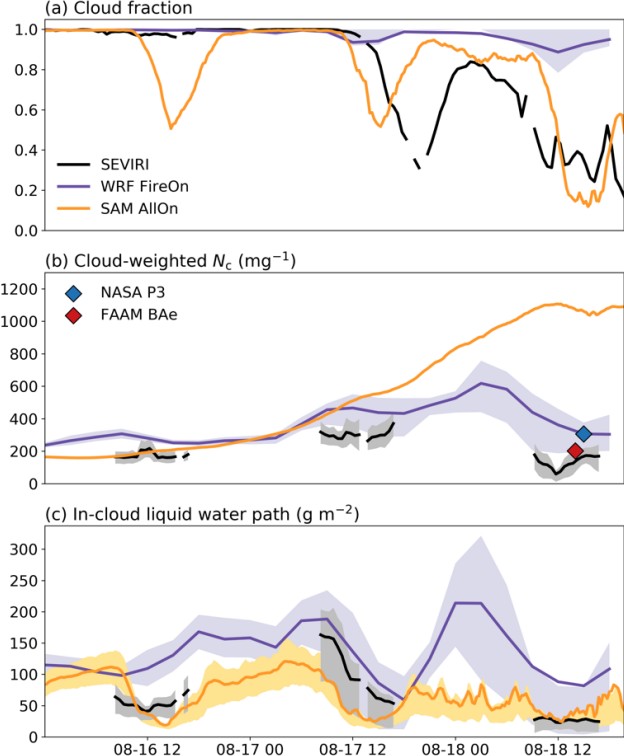

**Figure 21. Comparison of SEVIRI retrievals and model output over the course of the Lagrangian trajectory. Lines represent mean values and shading interquartile ranges for (a) cloud fraction, (b) cloud droplet number concentration, and (c) in-cloud liquid water**

**path. The markers in (b) represent the average $N_c$ weighted by cloud water content for the ORACLES (blue) and CLARIFY (red) flights.**





## 6 Discussion and conclusions

The massive quantities of smoke produced by largely anthropogenic burning in southern Africa that overlies the southeast Atlantic Ocean each July to October makes the SEA region ideal for the study of aerosol-radiation and aerosol-cloud interactions in many respects. Recent field studies, especially the intensive sampling of the area surrounding Ascension Island by ORACLES, CLARIFY, and LASIC on 18 August 2017, present a promising setup for a modeling case study of how smoke influences the stratocumulus-to-cumulus transition in the region. In this paper, we used such a case to force a large eddy simulation model with free tropospheric profiles from regional climate model output including and excluding selected smoke effects.

The WRF-CAM5 regional model output produces non-intuitive differences in the free troposphere above the MBL, with an expected warming (from smoke absorption) in the core of the smoke plume but "banding" features of cooler temperatures and strong moistening at plume top and weak warming and drying at the plume base. These effects are caused by the vertical displacement of the (formerly) continental airmass due to a reduction in large-scale subsidence driven by smoke radiative heating.

There is a large increase in cloud cover in SAM when all smoke effects are included. This is driven by large-scale thermodynamic and dynamic semi-direct effects, not by microphysics, and is partially counteracted by local below-cloud semi-direct effects. The importance of the subsidence both for plume location and for cloud evolution directly is a particularly striking result, especially as previous LES studies could not incorporate subsidence changes due to the limitation of using a small domain without the ability to interact with larger scales.

All but one of the SCTs simulated in the SAM cases followed the classical model of an entrainment-driven "deepening-warming" transition (Bretherton and Wyant, 1997; Wyant et al., 1997), rather than the more recently proposed model of a precipitation-driven "drizzle-depletion" transition (Yamaguchi et al., 2017). However, the largest radiative impact (in terms of the difference from the base AllOn case in Fig. 18) came from $AllOff_{N/2}$, which unlike the standard AllOff case did experience a runaway precipitation-scavenging feedback and only differed from AllOff in terms of its aerosol forcing. Importantly, the drizzle-driven transition is not only faster than that driven by entrainment but also differs qualitatively, with an end state more akin to shallow open-cell stratocumulus than deeper but scattered trade cumulus. There is some observational evidence that qualitatively different closed-cell to disorganized convection and closed-to-open-cell transitions can both occur under relatively similar large-scale meteorological conditions (Eastman et al., 2021). A more precise understanding of the factors favoring one transition mechanism over the other and which is more prevalent in reality (and whether the prevalence may change with projected future aerosol emission trajectories) would be a fruitful avenue for further inquiry.

One weakness of the WRF-CAM5 simulations is the necessity of parameterizing turbulent MBL processes that can be resolved directly in SAM (at least at the "large eddy," 10s-of-meters scale). The excessive low cloud coverage in all WRF simulations over essentially the entire oceanic domain, and the marked underestimate in other formulations of WRF and in other models as documented by Doherty et al. (2022), is a likely consequence of this deficiency — it is very difficult for





models that cannot directly resolve boundary layer motions to accurately represent stratocumulus clouds. The cloud fraction differences between the SAM simulations therefore give more insight into aerosol-driven radiative effects than those differences (or the lack thereof) between the WRF simulations. At the same time, the larger, regional-scale domain in WRF is able to capture large-scale smoke-circulation interactions that would not be possible to resolve in the relatively tiny SAM domain. Given the ultimate importance of semi-direct effects driven in large part by smoke/ex-CBL vertical shifts, the

impossibility of capturing those effects in SAM is a major limitation of relying on LES alone. The combination of the two modeling frameworks (by forcing SAM with the WRF output and nudging the free troposphere) thus combines the relative strengths of each.

       A limitation of both models being only run for several days is that there is insufficient time for a strong sea surface temperature response to emerge. Overlying smoke should produce a strong shading effect and cloud radiative effects (at least

in SAM) were more negative with the inclusion of smoke, so sea surface temperatures should decrease due to biomass burning effects, which would then increase the strength of the MBL inversion in general and thus cloudiness over the southeast Atlantic. Longer running regional or global climate models do simulate such SST effects (Mallet et al., 2020; Sakaeda et al., 2011). It is also worth keeping in mind that these results are for one particular case, and given the complexity of the plume transport, they may not generalize to other cases in the SEA.

We find that above-cloud humidity is an important factor driving the differences between LES simulations with and without smoke effects, especially toward the end of the transition. Moisture effects also proved important in the studies of Y15 and Z17, although for somewhat different reasons. In Y15, the increase in aerosol and decrease in precipitation was sufficient to maintain overcast conditions toward the end of the SCT simulation, whereas in our simulations (at least those based primarily on the FireOn forcing) the effect of aerosol on cloud fraction was limited compared to the effect of moisture. Moisture paired

with smoke in Y15 did, however, lead to a large increase in liquid water path toward the end of their simulation whereas in our simulations the moisture mainly acted to maintain a thin stratocumulus layer. Before contacting the cloud tops, elevated water vapor in Y15 mainly acted to suppress longwave radiative cooling at cloud top, TKE, and thus cloud fraction and MBL depth. The differences in the total amount of elevated water between the WRF FireOn, FireOff, and RadOff cases before plume (ex-CBL) contact is small both compared to the differences right at cloud top later in the runs and to the water vapor differences

in Y15. Similar to Y15, Z17 found that elevated water vapor causes cloud breakup via its radiative effect but entrainment of moister air increases liquid water path. The effect of moisture on cloud cover in Z17 was relatively mild compared to the large changes in our simulations.

       The moisture "banding" effect in the LES simulations poses a conceptual challenge as a "smoke" effect. Even if there were no moisture enhancement in the heart of the plume (a small enhancement was simulated in WRF-CAM5), there would

have been a large moisture effect due to the vertical displacement of the ex-CBL alone. Air sourced near the surface would almost always be expected to be moister than air subsiding after undergoing deep convection in the tropics or Southern Ocean. Given the negligible differences between the FireOff and RadOff moisture fields and the limitations of the representation of fires in WRF-CAM5 (only emissions from QFED, which do not include water vapor, are included), smoke-radiation



interactions must account for essentially all of the moisture effects in the models. Even the elevated moisture within the smoke plumes seen in reality (e.g., Adebiyi et al., 2015; Pistone et al., 2021) cannot plausibly be emitted from the burning vegetation itself. A very high-end estimate (characteristic of trees) for the ratio of water molecules emitted to carbon dioxide molecules from burning vegetation of 3-4 (Parmar et al., 2008) would require a 2 g/kg enhancement of water vapor (~3000 ppm) within the ex-CBL to be accompanied by an enhancement in carbon dioxide of ~800-1000 ppm. A more realistic "savannah grass" ratio of 1.25 (Parmar et al., 2008) would require ~2500 ppm of $CO_2$. No $CO_2$ concentration of that magnitude, or anything close, has been observed over the southeast Atlantic. For all three ORACLES deployments, the vast majority of $CO_2$ concentrations were measured as between 400 and 460 ppm and there were no measurements above 500 ppm.

Perhaps because of its timing (toward the end of the trajectory), the cool "band" at plume top dominated the cloud fraction difference between the SAM AllOn and TOff cases. Thus, the "large-scale" semi-direct effects caused by subsidence changes and the resulting cool/moist "band" from plume displacement wound up influencing the SCT more than the traditionally considered FT warming/inversion strengthening effect. Even the more "local" indirect and below-cloud semi-direct effects are modulated by the large-scale circulation changes because of their influence on plume location and therefore the strength, onset, and duration of smoke entrainment.

One final implication of these simulations is the inherent difficulty of disentangling different aerosol effects, at least when absorbing aerosols are involved. For instance, the difference in $N_c$ between the WRF-CAM5 FireOn and RadOff cases can accurately be described as an aerosol-cloud interaction effect caused by rapid adjustments to aerosol-radiation interactions, and any resulting cloud adjustments as rapid adjustments to aerosol-cloud interactions caused by rapid adjustments to aerosol-radiation interactions. Should these effects be classified as semi-direct effects or indirect effects or both? The potential convolution of direct and indirect effects is also a possible issue with observational analyses over the SEA. Changes in radiative effects with column aerosol measures could be simply measuring the aerosol increase for constant cloud properties, rather than decreasing cloud brightness, as inferred for the SEA in some previous work (Douglas and L'ecuyer, 2019). An increase in cloud albedo will also have the partially compensating effect of making the aerosol direct effect more positive. Even in models, diagnosing cloud radiative effect changes is complicated by the fact that the aerosol direct effect changes sign in clear versus cloudy skies, so comparisons of clear (not pristine) sky and all-sky fluxes include both the effects of clouds themselves and their influence on the direct radiative effect. Idealized modeling studies, particularly with clear counterfactuals (e.g., FireOff) and denial-of-mechanism attributes (such as the inhibition of aerosol-radiation interactions in this study), in conjunction with observational analyses, are a promising method of disentangling these effects and inferring causality in the SEA region and for other settings in which multiple competing aerosol effects co-occur.

**Code availability**

WRF is available from the University Corporation for Atmospheric Research following the instructions at https://www2.mmm.ucar.edu/wrf/users/download/get_sources_new.php and the wrf-python analysis package at https://wrf-



python.readthedocs.io/en/latest/. SAM is available from Marat Khairoutdinov at http://rossby.msrc.sunysb.edu/~marat/SAM.html.

**Data availability**

The model output and forcings created for this study are publicly available from the NOAA Chemical Sciences Laboratory's

Clouds, Aerosol, & Climate program at https://csl.noaa.gov/groups/csl9/datasets/ (Note: this is in progress; we can make output available to reviewers before the site is live, if requested). All ORACLES-2017 flight data are publicly available from the NASA Earth Science Project Office Data Archive (Oracles Science Team, 2020) at https://doi.org/10.5067/Suborbital/ORACLES/P3/2017_V2. SEVIRI data are available from the NASA Ames Research Center at https://cloud1.arc.nasa.gov/oracles/data/. CLARIFY flight data is publicly accessible from the UK Centre for Environmental

Data Analysis (CEDA) Archive (Facility for Airborne Atmospheric Measurements et al., 2017) at http://catalogue.ceda.ac.uk/uuid/38ab7089781a4560b067dd6c20af3769. LASIC data is available from the ARM Climate Research Facility Data Archive at https://www.arm.gov/research/campaigns/amf2016lasic. The authors gratefully acknowledge the NOAA Air Resources Laboratory (ARL) for the provision of the HYSPLIT transport and dispersion model used in this publication (https://www.ready.noaa.gov/HYSPLIT.php). MERRA-2 meteorological reanalysis data are publicly

available from NASA's Goddard Earth Sciences Data and Information Services Center (https://disc.gsfc.nasa.gov/).

**Author contributions**

MSD, PES, and RW conceptualized the study. PES ran WRF-CAM5 and MSD ran HYSPLIT and SAM with the assistance of GF, JK, and TY. MSD acquired all data for the ORACLES and CLARIFY campaigns and for SEVIRI and JZ acquired data for the LASIC campaign. MSD performed the formal analysis, visualization, and writing of the original draft manuscript. All

1085 authors edited the manuscript and provided feedback on the interpretation of results.

**Competing interests**

Paquita Zuidema is a guest editor for the ACP Special Issue: "ACP special issue: New observations and related modelling studies of the aerosol–cloud–climate system in the Southeast Atlantic and southern Africa regions" The rest of authors declare that they have no conflict of interest.

**Special issue statement**

This article is part of the special issue "New observations and related modelling studies of the aerosol–cloud–climate system in the Southeast Atlantic and southern Africa regions (ACP/AMT inter-journal SI)". It is not associated with a conference.



**Acknowledgements**

We thank the ORACLES, CLARIFY, and LASIC teams for their significant efforts of in collecting and making available data from the remote southeast Atlantic Ocean. Computing resources were provided by the Earth's Radiation Budget program through the National Oceanic and Atmospheric Administration's Climate Program Office (grant #03-01-07-001). We thank Peter Blossey, Christopher Bretherton, and Matthew Wyant for helpful advice and discussions.

**Financial support**

ORACLES is a NASA Earth Venture Suborbital- 2 investigation, funded by NASA's Earth Science Division and managed through the Earth System Science Pathfinder Program Office (grant no. NNH13ZDA001N-EVS2). MSD was supported by NASA headquarters under the NASA Earth and Space Science Fellowship Program (grant NNX-80NSSC17K0404) and by the CIRES Visiting Fellows Program that is funded by the NOAA Cooperative Agreement with CIRES (grant NA17OAR4320101). PZ and JZ acknowledge support from Department of Energy (DOE) ASR (grant DE-SC0021250); PES and PZ acknowledge additional support from DOE (grant DE-SC0018272). JZ additionally acknowledges support from a National Research Council Research Associateship Award. HG acknowledges support from the NASA ROSES program (grant 80NSSC21K1344).

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
