# Peer review of "Cloud adjustments from large-scale smoke-circulation interactions strongly modulate the southeast Atlantic stratocumulus-to-cumulus transition"

_Atmospheric Chemistry and Physics, 2022_

## Referee Comment (RC1)

**Review of "Cloud adjustments from large-scale smoke-circulation interactions strongly modulate the southeast Atlantic stratocumulus-to-cumulus transition" by Diamond et al.**

This work investigated the transition from stratocumulus to cumulus clouds in a Lagrangian way by a large eddy simulation model SAM with inputs from regional model WRF-CAM5. The authors leveraged the aircraft and ground-based observations in combination with model simulations to show that vertical displacement of the former continental boundary is responsible for the main variation in the free troposphere. This paper highlights the thermodynamic and dynamic feedback due to BB aerosol radiation effect as the most important factor in the transition from stratocumulus to cumulus clouds. This is an exceptionally thorough and convincing paper that is well written and presented. I only have a few minor comments that need to be addressed, thus recommending publication subject to minor revision.

**Minor**

1. Line 124-127. Can the author explain more why precipitation scavenges can result in a lower number of larger droplets?

2. Line 141-145. The authors suggest that BB aerosol in the African region is mainly due to anthropogenic agriculture burning. However, satellite images show that most of the burning is in the Savannah region (Figure 8 in Che et al., 2022, https://doi.org/10.5194/acp-2022-160). Is this consistent with the authors' claim?

3. Line 260-216. The authors used the high-resolution meteorological field from WRF-CAM to run HYSPLIT trajectories initialized from Ascension Island, but used the coarser GDAS data to run HYSPLIT trajectories at 2 km. Do different datasets have an impact on the results of backward trajectories? Why is a high-resolution weather field not chosen for 2km trajectories?

4. Line 636-638. The moist FT is considered to be due to ex-CBL. Is it possible that it is also due to BB aerosol heating of the FT, resulting in enhanced evaporation of the underlying cloud droplets?

5. Line 846-847. The TKE is similar for all cases, but why does Toff have a higher boundary layer height and Woff have a lower one than AllOn? Can the auther explain more?

---

## Author Response (AR1)

We would like to thank the reviewer, Dr. Che, for their constructive comments. Please find a detailed response that outlines our changes in response to the suggestions below.

*"1. Line 124-127. Can the author explain more why precipitation scavenges can result in a lower number of larger droplets?"*

This was originally phrased unclearly; clarified to read "leading to clouds with lower cloud droplet number concentrations and larger droplet sizes and thus greater precipitation".

*"2. Line 141-145. The authors suggest that BB aerosol in the African region is mainly due to anthropogenic agriculture burning. However, satellite images show that most of the burning is in the Savannah region (Figure 8 in Che et al., 2022, https://doi.org/10.5194/acp-2022-160). Is this consistent with the authors' claim?"*

Our understanding — based on the holiday effects (Earl et al., 2015; Pereira et al., 2015), daily cycle of burn intensity (Roberts et al., 2009), and long-term trends driven by land-use change (Andela et al., 2017) — is that much of the BB aerosol in the southern African region is anthropogenic in origin. We are not aware of any reason why the occurrence of fires primarily in savannah land types (as classified by the MODIS land cover product) would be inconsistent with a strong anthropogenic influence. However, as the exact breakdown between "natural" and anthropogenic fires is not directly related to our results or analysis, we've toned down some of the language that suggested anthropogenic influence was the majority of burning (which may be true, but would need to be quantified) to instead say that anthropogenic influence is large.

*"3. Line 260-216. The authors used the high-resolution meteorological field from WRF-CAM to run HYSPLIT trajectories initialized from Ascension Island, but used the coarser GDAS data to run HYSPLIT trajectories at 2 km. Do different datasets have an impact on the results of backward trajectories? Why is a high-resolution weather field not chosen for 2km trajectories?"*

We ran the 2 km trajectories using the system in place for in-field forecasting, whereas the back trajectory used to set up the LES case was run specially using the WRF-CAM5 wind fields. There may be slight discrepancies between the WRF-CAM5 and GDAS wind fields (although it is worth noting that WRF-CAM5 was initialized with NCEP reanalysis fields that should be consistent with GDAS); however, as we are only using the 15 August 2017 data as a loose constraint with a relatively large tolerance for "matches" (within a degree of longitude on either side of 5° E from 07:00 to 18:00 UTC), we doubt that the greater precision from using the WRF-CAM5 wind fields would have any bearing on our conclusions.

*"4. Line 636-638. The moist FT is considered to be due to ex-CBL. Is it possible that it is also due to BB aerosol heating of the FT, resulting in enhanced evaporation of the underlying cloud droplets?"*

Because the "ex-CBL" air contains significant moisture in all cases (FireOn, FireOff, and RadOff), and two of those cases do not contain any FT heating due to smoke, we are confident that the moisture is not primarily a semi-direct effect. There is some apparent enhancement of moisture in FireOn above that of FireOff and RadOff, which could plausibly represent an absorption effect. Enhanced evaporation of underlying cloud droplets cannot explain this difference, however, both because we are using total water mixing ratio (so the partitioning between vapor and liquid water is irrelevant) and because FT air is entrained into the growing MBL but detrainment of MBL air into the FT is minimal, and certainly negligible at altitudes of ~2-3 km.

*"5. Line 846-847. The TKE is similar for all cases, but why does Toff have a higher boundary layer height and Woff have a lower one than AllOn? Can the auther explain more?"*

We argue that the weaker inversion in TOff (greater subsidence in WOff) primarily explains the higher (lower) boundary layer heights as compared to AllOn even though TKE values are similar. We have rearranged portions of the text to clarify.

"TOff grows more rapidly than AllOn (Figure 16b), despite similar MBL turbulence (Figure 16h) and updraft speeds, due to the weaker inversion during most of the simulation, which also leads to enhanced entrainment of smoke and thus greater $N_c$ (Figure 16c)."

"The greater subsidence in WOff (Figure 6e) suppresses MBL growth by several hundred meters compared to AllOn (Figure 16b) despite similar TKE values (Figure 16h), resulting in less smoke entrainment and lower $N_c$ as well (Figure 16c-d)."

We would like to thank the reviewer for their constructive comments. Please find a detailed response that outlines our changes in response to the suggestions below.

*"Representativeness of simulated conditions. The SAM simulation AllOff_N/2 shows that if the FT aerosol concentrations are halved then the baseline cloud evolution is very different. This would imply that the response of the cloud to smoke is possibly sensitive to the baseline. The low cloud fraction comparison between SEVIRI and WRF over the domain (Figure S5) suggests to me that the observed SCT is better reproduced assuming a cleaner FT – is this true? If so, then this would suggest the bulk of results and conclusions of the paper are not representative of the region. To an extent the authors have acknowledged this on lines 993-995, but it would be interesting to hear how confident the authors are that the SAM simulations are representative of the mean state of the region."*

AllOff$_{N/2}$ does not reproduce the observations well — the MBL is much too shallow and $N_c$ much too low as compared to SEVIRI for the Lagrangian case. In the new Figure S6 (see below), it is also clear that the $O(10$ $mg^{-1})$ $N_c$ values from AllOff$_{N/2}$ are also not representative of the broader region. We do know that ultra-clean MBLs like those simulated in AllOff$_{N/2}$ are observed occasionally during the biomass burning season (Pennypacker et al., 2020), although they are not the mode. We have added a note about this in the conclusion: "Although the AllOff$_{N/2}$ simulation poorly matches the observations for this case, ultra-clean boundary layers are known to occur occasionally around Ascension Island during the biomass burning season (Pennypacker et al., 2020)."

We would hesitate to state that our results are representative of "the mean state of the region," and indeed would argue that the heterogeneity of smoke/ex-CBL influence makes the concept of a single representative mean state problematic for this region. We have also elaborated on this point in the conclusion: "It is also worth keeping in mind that these results are for one particular case, and given the complexity of the plume transport, they may not generalize to other cases in the SEA. In particular, the properties of the ex-CBL in the absence of biomass burning are fundamentally non-observable (at least with present land-use conditions) and thus are a major source of uncertainty, especially in terms of the aerosol background concentration. Airmass properties observed outside of the biomass burning season could offer some useful information, but seasonal differences preclude a direct comparison."

Pennypacker, S., Diamond, M., & Wood, R. (2020). Ultra-clean and smoky marine boundary layers frequently occur in the same season over the southeast Atlantic. *Atmospheric Chemistry and Physics, 20*(4), 2341-2351. doi:10.5194/acp-20-2341-2020

*"Enhanced moisture above the inversion. I think the banding effect is a really interesting result and adds to the growing appreciation of moist layers over the region and their role in cloud evolution. The authors discuss the moisture effects etc in lines 1028 to 1041 but I feel there are remaining questions that could be answered here. As pointed out, several studies (e.g.,*

*Adebiyi et al 2015; Pistone et al 2021) have observed high moisture content in elevated plumes of smoke – do the authors believe that the WRF simulation accurately reproduces the degree of moisture enhancement in the plume? If not, what impact do the authors think this would have on the results? Would we expect to see a stronger or buffered cloud response?"*

The moisture enhancement of a few (~3-5) g/kg in the ex-CBL in FireOn, FireOff, and RadOff as compared to other FT air is broadly in line with the observations from Adebiyi et al. (2015) Fig. 2, Pistone et al. (2021) Figs. 3 and 15, and Zhang & Zuidema (2021) Figs. 7-9. We have clarified this point in Section 3.2: "The moisture enhancement in the neutrally stratified layer in all three cases is similar in magnitude to that observed within smoke plumes over the southeast Atlantic (Adebiyi et al., 2015; Pistone et al., 2021; Zhang & Zuidema, 2021)."

Our results confirm that moisture changes on the order of a few g/kg can have large impacts on cloud development. Therefore, if WRF had simulated very different moisture enhancements, we agree this could have been an important source of uncertainty.

*"Line 328. Does turning off the shallow convection scheme impact the ability to reproduce the SCT?"*

We chose the model configuration with the shallow convection scheme turned off because the configuration with the scheme turned on had greatly suppressed cloud fraction and boundary layer depths whereas the configuration with scheme turned off performed well as compared to other models in terms of matching the ORACLES observations in Shinozuka et al. (2020) and Doherty et al. (2022). Although turning off the shallow convection hinders our ability to represent the SCT in terms of cloud fraction changes, we are able to capture an overall realistic boundary layer height evolution (Fig. 7b and 19) and reasonable changes in other cloud properties. Leaving the shallow convection scheme on would have hindered our ability to capture the evolution in terms of boundary layer evolution and thus likely in terms of cloud properties as well. Additionally, the cloud fraction suppression with the scheme on could be as problematic as the cloud fraction enhancement is with the scheme off.

*"Line 396. SAM not defined."*

Thanks for catching that — fixed.

*"Line 400. 'The LES is only nudged in the FT…' I suggest this is moved higher up in this or the previous paragraph."*

Fixed.

*"Line 408. I suggest introducing the model before the LES forcing section."*

We have removed the SAM-specific language in Section 2.3.1, as per the previous reviewer comment. This change means that the LES forcing section is now applicable to any LES (or single-column) model, including non-SAM models that may wish to use the forcing files we provide at https://csl.noaa.gov/groups/csl9/datasets/. We therefore prefer to introduce SAM after the LES forcing section, as the LES forcing is designed to be generic and not only compatible with SAM.

*"Line 443. Is this assumption appropriate? Assuming the wind-dependent source is sea salt wouldn't you expect a substantially higher hygroscopicity?"*

This assumption is not ideal, as the optical and chemical properties of smoke and sea salt differ greatly. However, we do not expect this to affect our results, as the FT source is much greater than the surface source in our simulations. We have added new text clarifying this: "This simplification is unlikely to affect our results, as the surface aerosol source is very small compared to the FT source in our simulations."

Additionally, even with the relatively non-hygroscopic smoke particles in our simulations, we see activation fractions approaching 1. Although these activation fractions are quite likely unrealistic, they suggest that better accounting for aerosol type in our current model setup would have negligible impacts on the simulation results.

*"Line 480 (and table 1). The use of 'aerosol-radiation interactions' and 'smoke-radiation interactions' is confusing. By referring to 'aerosol-radiation interactions' are you essentially referring to any non-smoke aerosol? Please clarify. This would avoid the confusion on line 480 which sounds like semi-direct effects in the MBL are negligible."*

A clarification has been included in the Table 1 caption ("Aerosol" refers to all aerosol species whereas "smoke" refers specifically to the aerosol attributable to the QFED fire emissions) and language has been updated elsewhere (e.g., " Aerosol-radiation interactions are represented with a lookup-table method" instead of "Smoke-radiation interactions" in original Line 427). The MBL semi-direct effects in the test "AllOff"-like case including aerosol-radiation interactions were indeed negligible as compared to those between AllOn and AsiAbs, mostly because the aerosol concentration is an order of magnitude lower than in AllOn.

*"Line 569 (and line 752). How do the WRF simulations compare to the SEVIRI Nc maps?"*

Good question! We've added a new Supporting Information Figure S6 comparing SEVIRI $N_c$ with the WRF simulations and have updated the text in Section 3.2: "The peak values within the smoke-affected area are much higher in WRF-CAM5 FireOn and RadOff than retrieved by passive satellite sensors (Fig. S6; see also climatological values in Grosvenor et al., 2018). Surprisingly, the FireOff $N_c$ values most closely match the SEVIRI observations, although this may be a coincidence from a general overestimate of aerosol number concentrations in WRF-CAM5 or retrieval biases affecting SEVIRI and other passive satellite sensors (Grosvenor et al.,

2018; Meyer et al., 2015). The spatial distribution of $N_c$ in the Eulerian sense tracks the evolution of the FireOn and RadOff trajectories in general and is broadly consistent with the pattern observed from SEVIRI."

Additionally, there is currently a paper in preparation (Howes et al., *in prep*) that includes a detailed evaluation of WRF-CAM5, including comparisons with in-situ $N_c$.

*"Line 808. Could this be a saturation effect? If the model had more representative Na values in the MBL (Figure 19) might there be more sensitivity?"*

Figure 19 does not show $N_a$ values for FireOff (which is most relevant for the AllOff case), but the fact that AllOff$_{N/2}$ does show a drizzle-driven transition suggests that the transition is sensitive to drizzle when background aerosol concentrations are sufficiently low. The $N_a$ values in the MBL for FireOff are of O(100 mg$^{-1}$), however (see, e.g., Fig. 7d), which are similar to (and indeed lower than) the observations shown in Fig. 19. So we can safely conclude that forcing the model with concentrations more similar to the observations from 18 August 2017 (~500 mg$^{-1}$) would not lead to a drizzle-driven transition since the model with lower concentrations of ~100-200 mg$^{-1}$ didn't (AllOff and NOff).

We would like to thank the reviewer for their constructive comments. Please find a detailed response that outlines our changes in response to the suggestions below.

*"1. Why do the authors use different meteorological datasets to run trajectories in the boundary layer and free-troposphere in figure 3? Also, for the free-tropospheric trajectories, why was 2 km altitude chosen as a starting point i.e. above the boundary layer/middle of the aerosol plume etc. Noting that the observed aerosol plume closer to Ascension Island is at a higher altitude (Fig 2)."*

We ran the 2 km trajectories using the system in place for in-field forecasting, whereas the back trajectory used to set up the LES case was run specially using the WRF-CAM5 wind fields. There may be slight discrepancies between the WRF-CAM5 and GDAS wind fields (although it is worth noting that WRF-CAM5 was initialized with NCEP reanalysis fields that should be consistent with GDAS); however, as we are only using the 15 August 2017 data as a loose constraint with a relatively large tolerance for "matches" (within a degree of longitude on either side of 5° E from 07:00 to 18:00 UTC), we doubt that the greater precision from using the WRF-CAM5 wind fields would have any bearing on our conclusions.

2 km was chosen as the trajectory starting point to safely remain above the MBL while still being fairly representative of the aerosol entraining into the MBL. The higher-altitude aerosol plume near Ascension Island is not directly related to the smoke plume/ex-CBL that entrained into the MBL during our case and could not have contributed any aerosol to the MBL during our period of interest. We have added some clarifying text to Section 2.1: "2 km is chosen as the trajectory starting point to safely remain above the MBL while still being fairly representative of the aerosol entraining into the MBL."

*"2. WRF has a high bias in the FT aerosol number concentration, but the OC and BC mass loadings are comparable to those observed (Fig S1). Can the authors confirm that the model data in Fig 4a/Fig S1b is the accumulation mode only that this mode does represent a comparable size range as the aircraft measurement? If yes, are there significant concentrations of aerosol species in addition to OC and BC in the model that might account for the difference?"*

The model number concentration data in Figure 4a/Figure S1b are for the accumulation mode bin of the MAM3 scheme, in which the mode dry diameter ranges between 58 and 270 nm (10th to 90th percentiles) and the geometric standard deviation is set at 1.8 (Liu et al., 2012; their Table 1). The UHSAS data used is for dry diameters between 60 and 600 nm and the PCASP data is for dry diameters between 100 and 3,000 nm. The smoke mode is just under 200 nm and the vast majority of the number concentration is captured by MAM3 and both instruments. For mass, neither the model nor the observational data are constrained to represent the accumulation mode only.

Sulfate contributes to the aerosol number and mass concentrations in addition to OC and BC, although there are questions about to what extent the aerosol particles are internal or external mixtures in reality (they are internal in MAM3). Sulfate also doesn't differ strongly between WRF-CAM5 and the observations, at least during the 15 August 2017 flight. Work to better understand how the WRF-CAM5 aerosol size and mass distribution compares to observations and the reasons why they may differ is ongoing (Howes et al., *in prep*).

A clarification of the size range has been added to Section 2.3.1: "A modal aerosol module (Liu et al., 2012) with Aitken, accumulation [mode dry diameters from 58 to 270 nm (10th to 90th percentiles) and geometric standard deviation of 1.8], and coarse modes (MAM3) coupled with a gas phase chemistry scheme (Zaveri and Peters, 1999) is used for smoke (and other aerosol) properties. ($N_a$ from WRF-CAM5 refers to the accumulation mode only unless otherwise specified.)"

"3. Figure 5 uses altitude (km) and Fig 6 uses pressure (hPa) on the y-axis. Is it possible to have consistency so that features at different altitudes can be compared more easily?"

Changed to using altitude on the y-axis.

"4. Figure 7: Can the authors comment on why the trend in cloud drop concentration in the WRF model does not follow the MBL aerosol number concentration, particularly towards the end of the trajectory? This seems counterintuitive and is different to the results from the SAM model shown in figure 15."

The activation fraction in WRF decreases with increasing aerosol loading, which is reasonable behavior for highly polluted MBLs (Kacarab et al., 2020). MBL-average TKE, and thus updraft velocities, decline over time in WRF as well, which can further explain decreasing activation fractions. The SAM activation fractions (near unity even in very polluted conditions) may be the unrealistic ones. A discussion was added to Section 3.1: "In all cases, the fraction of aerosol activating to form cloud droplets decreases over time as the MBL becomes more polluted (Figure 7d) and MBL-average TKE and thus updraft velocities decline (Figure 7h)."

"5. Figure 8 d: "$\Theta_{l,FireOn} < \Theta_{l,FireOff}$ at plume top" is only strictly true if one considers the plume top height being fixed at the height of the FireOn simulation. $\Theta_l$ at the plume top height of the FireOff/RadOff simulations is actually cooler than the value at the plume top height of the FireOn simulation."

We have now clarified in the caption that "plume top" in the figure refers to the FireOn plume top: ""Plume top," "in plume," and "below plume" here refer to the location of the FireOn ex-CBL."

"6. Figure 14: Can the authors change the colour bar for cloud water mixing ratio, given that it uses similar colours to the cloud droplet concentration shading on which it is overlaid."

Changed to purple instead of blue.

*"7. Line 878: Consider rephrasing "the effect of subsidence exceeds that of the indirect effect", given that this is only when comparing against the baseline aerosol case. Arguably, the most dramatic impact on the cloud field does result from microphysics i.e. in the simulation that reduces the FT aerosol concentration by a factor of 2."*

Changed to: "The effect of subsidence exceeds that of the indirect effect in this study except in the case of extremely low background aerosol concentration."

*"8. Section 5: Refer the reader to Barrett et al. (2022) for a more complete description of the joint flight."*

Done. "The reader is referred to Barrett et al. (2022) for a more complete description of the joint flight."

*"9. Line 910: Can the authors also comment on possible reasons for the cool and moist bias in the MBL in the SAM simulation, when compared to the observations."*

This is likely related to the cool/moist biases in the entraining FT air. We have added text in Section 5: "The cooler and moister MBLs simulated in WRF-CAM5 and SAM also may be related to entrainment of cooler and moister FT air than in reality due to the prolonged plume/ex-CBL presence."

*"10. Line 913: Although the regional simulation is free-running, I would have thought that the aerosol plume takes several days to reach Ascension Island from the source region, as can be inferred from the trajectories in figure 3. Assuming that the model was initialised with a realistic aerosol location/amount near the source region, then does this point to model errors in the aerosol transport further offshore? Have the authors made any comparisons with satellite measurements of the plume location throughout the simulation period e.g with something like CALIOP/CATS/MODIS above cloud AOD/SEVIRI above cloud AOD etc, in order to examine this?"*

Freshly-emitted smoke on the continents does take several days to reach the offshore cloud areas of interest here. There is pre-existing smoke over the southeast Atlantic at the 14 August 2017 point of reinitialization that came from the previous WRF-CAM5 runs. Aerosol is only initialized once from CAMS in July and then evolves freely afterward (with CAMS continuing to provide boundary conditions). We have added a clarification of this point in Section 2.3.1: "Smoke already over the southeast Atlantic at the start of the reinitialization for the FireOn and RadOff simulations comes from the previous WRF-CAM5 initializations and new smoke is generated over the continent based on the QFED emissions."

We did roughly compare plume properties/location with CALIOP. There are matches with qualitative agreement between the CALIOP plume location and WRF-CAM5 toward the beginning of the trajectory. The model evaluations in Yohei et al. (2020) and Doherty et al. (2022) also support WRF-CAM5 simulating realistic smoke transport.

*"11. Line 995: It may be worth re-stating here, that the simulations that followed the classical model of an entrainment driven transition had much higher aerosol loadings in both the free troposphere and in the boundary layer than were measured by the ground site and aircraft. Suggesting that they may not be representative of typical conditions in this part of the SE Atlantic e.g. Fig 20."*

The statement that the simulations following the entrainment-driven transition occurred for higher aerosol loadings than that observed is not true for the NOff and AllOff cases, which have aerosol loadings much lower than observed (as expected based on the exclusion of smoke particles). We have added WRF FireOff and SAM AllOff and AllOff$_{N/2}$ values to Figure 20 and additional text in Section 5: "$N_a$ from the WRF FireOff and SAM AllOff runs are well below the LASIC observations on 18 August 2017 but within the range of values typically observed during August 2017 more generally (Pennypacker et al., 2020). Aerosol concentrations from the ultra-clean SAM AllOff$_{N/2}$ simulation are substantially lower than what was typically observed during August 2017 but are still well within the observed range. The range of aerosol conditions from our simulations therefore span the range of observations from LASIC relatively well."

*"12. Given the above point, and that the strongest cloud response from the baseline case arguably arises from the SAM simulation that is initialized with a factor of 2 reduction in the free-tropospheric aerosol concentrations (which is also perhaps in better agreement with observations – Fig S1), I do wonder if the title of the paper should be adjusted. As it appears from the results presented, that microphysical controls can be as important in modulating the SCT as the semi-direct effects (large-scale smoke-circulation interactions) that result from heating within the free-tropospheric aerosol plume."*

We should emphasize that AllOff$_{N/2}$ has a factor of 2 reduction as compared to the smoke-free case, not the case with smoke included that should be most representative of real-world conditions. The AllOff and NOff cases have aerosol concentrations much lower than that observed and still had entrainment, not drizzle, driven transitions. While we completely agree with the statement that microphysics can be the dominant driver of the transition under the right conditions, our novel contribution in the present work relates much more closely to the large-scale semi-direct effects, which have not previously been simulated in an LES model. The fact that drizzle-driven transitions can occur is reinforced by our AllOff$_{N/2}$ case but was previously shown by Yamaguchi et al. (2015, 2017) and is now the subject of an LES intercomparison study (including many of the current authors) that will explore this question in greater detail than was possible in the present work. We have more explicitly acknowledged this ongoing work in the conclusion: "A more precise understanding of the factors favoring one

transition mechanism over the other and which is more prevalent in reality (and whether the prevalence may change with projected future aerosol emission trajectories) would be a fruitful avenue for further inquiry and is the subject of ongoing work."